# Understanding the Role of Momentum in Stochastic Gradient Methods

**Igor Gitman**      **Hunter Lang**      **Pengchuan Zhang**      **Lin Xiao**

Microsoft Research AI
Redmond, WA 98052, USA
{igor.gitman, hunter.lang, penzhan, lin.xiao}@microsoft.com

## Abstract

The use of momentum in stochastic gradient methods has become a widespread practice in machine learning. Different variants of momentum, including heavy-ball momentum, Nesterov's accelerated gradient (NAG), and quasi-hyperbolic momentum (QHM), have demonstrated success on various tasks. Despite these empirical successes, there is a lack of clear understanding of how the momentum parameters affect convergence and various performance measures of different algorithms. In this paper, we use the general formulation of QHM to give a unified analysis of several popular algorithms, covering their asymptotic convergence conditions, stability regions, and properties of their stationary distributions. In addition, by combining the results on convergence rates and stationary distributions, we obtain sometimes counter-intuitive practical guidelines for setting the learning rate and momentum parameters.

## 1 Introduction

Stochastic gradient methods have become extremely popular in machine learning for solving stochastic optimization problems of the form

$$\underset{x \in \mathbb{R}^n}{\text{minimize}} \ F(x) \triangleq \mathbf{E}_\zeta \big[ f(x, \zeta) \big], \tag{1}$$

where $\zeta$ is a random variable representing data sampled from some (unknown) probability distribution, $x \in \mathbb{R}^n$ represents the parameters of a machine learning model (e.g., the weight matrices in a neural network), and $f$ is a loss function associated with the model parameters and any sample $\zeta$. Many variants of the stochastic gradient methods can be written in the form of

$$x^{k+1} = x^k - \alpha_k d^k, \tag{2}$$

where $d^k$ is a (stochastic) search direction and $\alpha_k > 0$ is the step size or learning rate. The classical stochastic gradient descent (SGD) [31] method uses $d^k = \nabla_x f(x^k, \zeta^k)$, where $\zeta^k$ is a random sample collected at step $k$. For the ease of notation, we use $g^k$ to denote $\nabla_x f(x^k, \zeta^k)$ throughout this paper.

There is a vast literature on modifications of SGD that aim to improve its theoretical and empirical performance. The most common such modification is the addition of a *momentum* term, which sets the search direction $d^k$ as the combination of the current stochastic gradient $g^k$ and past search directions. For example, the stochastic variant of Polyak's heavy ball method [26] uses

$$d^k = g^k + \beta_k d^{k-1}, \tag{3}$$

where $\beta_k \in [0, 1)$. We call the combination of (2) and (3) the Stochastic Heavy Ball (SHB) method. Gupal and Bazhenov [9] studied a "normalized" version of SHB, where

$$d^k = (1 - \beta_k)g^k + \beta_k d^{k-1}. \tag{4}$$

In the context of modern deep learning, Sutskever et al. [34] proposed to use a stochastic variant of Nesterov's accelerated gradient (NAG) method, where

$$d^k = \nabla_x f\left(x^k - \alpha_k \beta_k d^{k-1}, \zeta^k\right) + \beta_k d^{k-1}. \tag{5}$$

The number of variations on momentum has kept growing in recent years; see, e.g., Synthesized Nesterov Variants (SNV) [17], Triple Momentum [36], Robust Momentum [3], PID Control-based methods [1], Accelerated SGD (AccSGD) [12], and Quasi-Hyperbolic Momentum (QHM) [18].

Despite various empirical successes reported for these different methods, there is a lack of clear understanding of how the different forms of momentum and their associated parameters affect convergence properties of the algorithms and other performance measures, such as final loss value. For example, Sutskever et al. [34] show that momentum is critical to obtaining good performance in deep learning. But using different parametrizations, Ma and Yarats [18] claim that momentum may have little practical effect. In order to clear up this confusion, several recent works [see, e.g., 40, 1, 18] have aimed to develop and analyze general frameworks that capture many different momentum methods as special cases.

In this paper, we focus on a class of algorithms captured by the general form of QHM [18]:

$$
\begin{aligned}
d^k &= (1 - \beta_k)g^k + \beta_k d^{k-1}, \\
x^{k+1} &= x^k - \alpha_k \left[(1 - \nu_k)g^k + \nu_k d^k\right],
\end{aligned}
\tag{6}
$$

where the parameter $\nu_k \in [0, 1]$ interpolates between SGD ($\nu_k = 0$) and (normalized) SHB ($\nu_k = 1$). When the parameters $\alpha_k$, $\beta_k$ and $\nu_k$ are held constant (thus the subscript $k$ can be omitted) and $\nu = \beta$, it recovers a normalized variant of NAG with an additional coefficient $1 - \beta_k$ on the stochastic gradient term in (5) (see Appendix A). In addition, Ma and Yarats [18] show that different settings of $\alpha_k$, $\beta_k$ and $\nu_k$ recover other variants such as AccSGD [12], Robust Momentum [3], and Triple Momentum [36]. They also show that it is equivalent to SNV [17] and special cases of PID Control (either PI or PD) [1]. However, there is little theoretical analysis of QHM in general. In this paper, we take advantage of its general formulation to derive a unified set of analytic results that help us better understand the role of momentum in stochastic gradient methods.

## 1.1 Contributions and outline

Our theoretical results on the QHM model (6) cover three different aspects: asymptotic convergence with probability one, stability region and local convergence rates, and characterizations of the stationary distribution of $\{x^k\}$ under constant parameters $\alpha$, $\beta$, and $\nu$. Specifically:

- In Section 3, we show that for minimizing smooth nonconvex functions, QHM converges almost surely as $\beta_k \to 0$ for arbitrary values of $\nu_k$. And more surprisingly, we show that QHM converges as $\nu_k \beta_k \to 1$ (which requires both $\nu_k \to 1$ and $\beta_k \to 1$) as long as $\nu_k \beta_k \to 1$ slow enough, as compared with the speed of $\alpha_k \to 0$.

- In Section 4, we consider local convergence behaviors of QHM for fixed parameters $\alpha$, $\beta$, and $\nu$. In particular, we derive joint conditions on $(\alpha, \beta, \nu)$ that ensure local stability (or convergence when there is no stochastic noise in the gradient approximations) of the algorithm near a strict local minimum. We also characterize the local convergence rate within the stability region.

- In Section 5, we investigate the stationary distribution of $\{x^k\}$ generated by the QHM dynamics around a local minimum (using a simple quadratic model with noise). We derive the dependence of the stationary variance on $(\alpha, \beta, \nu)$ up to the second-order Taylor expansion in $\alpha$. These results reveal interesting effects of $\beta$ and $\nu$ that cannot be seen from first-order expansions.

Our asymptotic convergence results in Section 3 give strong guarantees for the convergence of QHM with diminishing learning rates under different regimes ($\beta_k \to 0$ and $\beta_k \to 1$). However, as with most asymptotic results, they provide limited guidance on how to set the parameters in practice for fast convergence. Our results in Sections 4 and 5 complement the asymptotic results by providing principled guidelines for tuning these parameters. For example, one of the most effective schemes used in deep learning practice is called "constant and drop", where constant parameters $(\alpha, \beta, \nu)$ are used to train the model for a long period until it reaches a stationary state and then the learning rate $\alpha$ is dropped by a constant factor for refined training. Each stage of the constant-and-drop scheme runs variants of QHM with constant parameters, and their choices dictate the overall performance of the algorithm. In Section 6, by combining our results in Sections 4 and 5, we obtain new and, in some cases, counter-intuitive insight into how to set these parameters in practice.

## 2 Related work

**Asymptotic convergence**   There exist many classical results concerning the asymptotic convergence of the stochastic gradient methods [see, e.g. 37, 28, 14, and references therein]. For the classical SGD method without momentum, i.e., (2) with $d^k = g^k$, a well-known general condition for asymptotic convergence is $\sum_{k=0}^{\infty} \alpha_k = \infty$ and $\sum_{k=0}^{\infty} \alpha_k^2 < \infty$. In general, we will always need $\alpha_k \to 0$ to counteract the effect of noise. But interestingly, the conditions on $\beta_k$ are much less restricted. For normalized SHB, Polyak [27] and Kaniovski [11] studied its asymptotic convergence properties in the regime of $\alpha_k \to 0$ and $\beta_k \to 0$, while Gupal and Bazhenov [9] investigated asymptotic convergence in the regime of $\alpha_k \to 0$ and $\beta_k \to 1$, both for *convex* optimization problems. More recently, Gadat et al. [7] extended asymptotic convergence analysis for the normalized SHB update to smooth nonconvex functions for $\beta_k \to 1$. In this work we generalize the classical SGD and SHB results to the case of QHM for smooth nonconvex functions.

**Local convergence rate**   The stability region and local convergence rate of the deterministic gradient descent and heavy ball algorithms were established by Boris Polyak for the case of convex functions near a strict twice-differentiable local minimum [29, 26]. For this class of functions heavy ball method is optimal in terms of the local convergence rate [21]. However, it might fail to converge globally for the general strongly convex twice-differentiable functions [17] and is no longer optimal for the class of smooth convex functions. For the latter case, Nesterov's accelerated gradient was shown to attain the optimal *global* convergence rate [22, 23]. In this paper we extend the results of Polyak [26] on local convergence to the more general QHM algorithm.

**Stationary analysis**   The limit behavior analysis of SGD algorithms with momentum and constant step size was used in various applications. [25, 39, 15] establish sufficient conditions on detecting whether iterates reach stationarity and use them in combination with statistical tests to automatically change learning rate during training. [6, 4] prove many properties of limiting behavior of SGD with constant step size by using tools from Markov chain theory. Our results are most closely related to the work of Mandt et al. [19] who use stationary analysis of SGD with momentum to perform approximate Bayesian inference. In fact, our Theorem 4 extends their results to the case of QHM and our Theorem 5 establishes more precise relations (to the second order in $\alpha$), revealing interesting dependence on the parameters $\beta$ and $\nu$ which cannot be seen from the first order equations.

## 3 Asymptotic convergence

In this section, we generalize the classical asymptotic results to provide conditions under which QHM converges almost surely to a stationary point for smooth *nonconvex* functions. Throughout this section, "a.s." refers to "almost surely". We need to make the following assumptions.

**Assumption A.** *The following conditions hold for F defined in* (1) *and the stochastic gradient oracle:*

1. *$F$ is differentiable and $\nabla F$ is Lipschitz continuous, i.e., there is a constant $L$ such that*

$$\|\nabla F(x) - \nabla F(y)\| \le L\|x - y\|, \qquad x, y \in \mathbb{R}^n.$$

2. *$F$ is bounded below and $\|\nabla F(x)\|$ is bounded above, i.e., there exist $F_*$ and $G$ such that*

$$F(x) \ge F_*, \qquad \|\nabla F(x)\| \le G, \qquad x \in \mathbb{R}^n.$$

3. *For $k = 0, 1, 2, \ldots$, the stochastic gradient $g^k = \nabla F(x^k) + \xi^k$, where the random noise $\xi^k$ satisfies*

$$\mathbf{E}_k[\xi^k] = 0, \qquad \mathbf{E}_k\left[\|\xi^k\|^2\right] \le C \quad a.s.$$

*where $\mathbf{E}_k[\cdot]$ denotes expectation conditioned on $\{x^0, g^0, \ldots, x^{k-1}, g^{k-1}, x^k\}$, and $C$ is a constant.*

Note that Assumption A.3 allows the distribution of $\xi^k$ to depend on $x^k$, and we simply require the second moment to be conditionally bounded uniformly in $k$. The assumption $\|\nabla F(x)\| \le G$ can be removed if we assume a bounded domain for $x$. However, this will complicate the proof by requiring special treatment (e.g., using the machinery of gradient mapping [24]) when $\{x^k\}$ converges to the boundary of the domain. Here we assume this condition to simplify the analysis.

By convergence to a stationary point, we mean that the sequence $\{x^k\}$ satisfies the condition

$$\liminf_{k \to \infty} \|\nabla F(x^k)\| = 0 \quad a.s. \tag{7}$$

Intuitively, as $\beta_k \to 0$, regardless of $\nu_k$, the QHM dynamics become more like SGD, so there should be no issue with convergence. The following theorem, which generalizes the analysis technique of Ruszczyński and Syski [33] to QHM, shows formally that this is indeed the case:

**Theorem 1.** *Let F satisfy Assumption A. Additionally, assume $0 \le \nu_k \le 1$ and the sequences $\{\alpha_k\}$ and $\{\beta_k\}$ satisfy the following conditions:*

$$\sum_{k=0}^{\infty} \alpha_k = \infty, \qquad \sum_{k=0}^{\infty} \alpha_k^2 < \infty, \qquad \lim_{k\to\infty} \beta_k = 0, \qquad \bar{\beta} \triangleq \sup_k \beta_k < 1.$$

*Then the sequence $\{x^k\}$ generated by the QHM algorithm* (6) *satisfies* (7). *Moreover, we have*

$$\limsup_{k\to\infty} F(x^k) = \limsup_{k\to\infty,\ \|\nabla F(x^k)\|\to 0} F(x^k) \quad a.s. \tag{8}$$

More surprisingly, however, one can actually send $\nu_k \beta_k \to 1$ as long as $\nu_k \beta_k \to 1$ slow enough, although we require a stronger condition on the noise $\xi$. We extend the technique of Gupal and Bazhenov [9] to show asymptotic convergence of QHM for minimizing smooth nonconvex functions.

**Theorem 2.** *Let F satisfy assumption A, and additionally assume that $\|\xi^k\|^2 < C$ almost surely, i.e., the noise $\xi$ is a.s. bounded. Let the sequences $\{\alpha_k\}$, $\{\beta_k\}$, and $\{\nu_k\}$ satisfy the following conditions:*

$$\sum_{k=0}^{\infty} \alpha_k = \infty, \qquad \sum_{k=0}^{\infty} (1 - \nu_k \beta_k)^2 < \infty, \qquad \sum_{k=0}^{\infty} \frac{\alpha_k^2}{1 - \nu_k \beta_k} < \infty, \qquad \lim_{k\to\infty} \beta_k = 1.$$

*Then the sequence $\{x^k\}$ generated by Algorithm* (6) *satisfies* (7).

The conditions in Theorem 2 can be satisfied by, for example, taking $\alpha_k = k^{-\omega}$ and $(1 - \nu_k \beta_k) = k^{-c}$ for $\frac{1+c}{2} < \omega \le 1$ and $\frac{1}{2} < c < 1$. We should note that, even though setting $\nu_k \beta_k \to 1$ is somewhat unusual in practice, we think the result of Theorem 2 is interesting from both theoretical and practical points of view. From the theoretical side, this result shows that it is possible to always be increasing the amount of momentum (in the limit when $\nu_k \beta_k = 1$, we are not using the fresh gradient information at all) and still obtain convergence for smooth functions. From the practical point of view, our Theorem 5 in Section 5 shows that for a fixed $\alpha$, increasing $\nu_k \beta_k$ might lead to smaller stationary distribution size, which may give better empirical results.

Also, note that when $\nu_k = \beta_k$, Theorems 1 and 2 give asymptotic convergence guarantees for the common practical variant of NAG, which have not appeared in the literature before. However, we should mention that the bounded noise assumption of Theorem 2 (i.e. $\|\xi^k\|^2 < C$ a.s.) is quite restrictive. In fact, Ruszczyński and Syski [32] prove a similar result for SGM with a more general noise condition, and their technique may extend to QHM, but bounded noise greatly simplifies the derivations. We provide the proofs of Theorems 1 and 2 in Appendix B.

The results in this section indicate that both $\beta_k \to 0$ and $\nu_k \beta_k \to 1$ are admissible from the perspective of asymptotic convergence. However, they give limited guidance on how to choose momentum parameters in practice, where non-asymptotic behaviors are of main concern. In the next two sections, we study local convergence and stationary behaviors of QHM with constant learning rate and momentum parameters; our analysis provides new insights that could be very useful in practice.

## 4 Stability region and local convergence rate

Let the sequence $\{x^k\}$ be generated by the QHM algorithm (6) with constant parameters $\alpha_k = \alpha$, $\beta_k = \beta$ and $\nu_k = \nu$. In this case, $x^k$ does not converge to any local minimum in the asymptotic sense, but its distribution may converge to a stationary distribution around a local minimum. Since the objective function $F$ is smooth, we can approximate $F$ around a strict local minimum $x^*$ by a convex quadratic function. Since $\nabla F(x^*) = 0$, we have

$$F(x) \approx F(x^*) + \tfrac{1}{2}(x - x^*)^T \nabla^2 F(x^*)(x - x^*),$$

where the Hessian $\nabla^2 F(x^*)$ is positive definite. Therefore, for the ease of analysis, we focus on convex quadratic functions of the form $F(x) = (1/2)(x - x^*)^T A(x - x^*)$, where $A$ is positive definite (and we can set $x^* = 0$ without loss of generality). In addition, we assume

$$g^k = \nabla F(x^k) + \xi^k = A(x - x^*) + \xi^k, \tag{9}$$

where the noise $\xi^k$ satisfies Assumption A.3 and in addition, $\xi^k$ is independent of $x^k$ for all $k \geq 0$. Mandt et al. [19] observe that this independence assumption often holds approximately when the dynamics of SHB are approaching stationarity around a local minimum.

Under the above assumptions, the behaviors of QHM can be described by a linear dynamical system driven by i.i.d. noise. More specifically, let $z^k = [d^{k-1}; x^k - x^*] \in \mathbb{R}^{2n}$ be an augmented state vector, then the dynamics of (6) can be written as (see Appendix E for details)

$$z^{k+1} = Tz^k + S\xi^k, \tag{10}$$

where $T$ and $S$ are functions of $(\alpha, \beta, \nu)$ and $A$:

$$T = \begin{bmatrix} \beta I & (1-\beta)A \\ -\alpha\nu\beta I & I - \alpha(1-\nu\beta)A \end{bmatrix}, \qquad S = \begin{bmatrix} (1-\beta)I \\ -\alpha(1-\nu\beta)I \end{bmatrix}. \tag{11}$$

It is well-known that the linear system (10) is stable if and only if spectral radius of $T$, denoted by $\rho(T)$, is less than 1. When $\rho(T) < 1$, the dynamics of (10) is the superposition of two components:

- A deterministic part described by the dynamics $z^{k+1} = Tz^k$ with initial condition $z^0 = [0; x^0]$ (we always take $d^{-1} = 0$). This part asymptotically decays to zero.
- An auto-regressive stochastic process (10) driven by $\{\xi^k\}$ with zero initial condition $z^0 = [0; 0]$.

Roughly speaking, $\rho(T)$ determines how fast the dynamics converge from an arbitrary initial point $x^0$ to the stationary distribution, while properties of the stationary distribution (such as its variance and auto-correlations) depends on the full spectrum of the matrix $T$ as well as $S$. Both aspects have important implications for the practical performance of QHM on stochastic optimization problems. Often there are trade-offs that we have to make in choosing the parameters $\alpha$, $\beta$ and $\nu$ to balance the transient convergence behavior and stationary distribution properties.

In the rest of this section, we focus on the deterministic dynamics $z^{k+1} = Tz^k$ to derive the conditions on $(\alpha, \beta, \nu)$ that ensure $\rho(T) < 1$ and characterize the convergence rate. Let $\lambda_i(A)$ for $i = 1, \ldots, n$ denote the eigenvalues of $A$ (they are all real and positive). In addition, we define

$$\mu = \min_{i=1,\ldots,n} \lambda_i(A), \qquad L = \max_{i=1,\ldots,n} \lambda_i(A), \qquad \kappa = L/\mu,$$

where $\kappa$ is the condition number. The local convergence rate for strictly convex quadratic functions is well studied for the case of gradient descent ($\nu = 0$) and heavy ball ($\nu = 1$) [26]. In fact, heavy ball achieves the best possible convergence rate of $(\sqrt{\kappa} - 1)/(\sqrt{\kappa} + 1)$[23]. Thus, it is immediately clear that the optimal convergence rate of QHM will be the same and will be achieved with $\nu = 1$. However, there are no results in the literature characterizing how the optimal rate or optimal parameters change as a function of $\nu$. Our next result establishes the convergence region and dependence of the convergence rate on the parameters $\alpha$, $\beta$, and $\nu$. We present the result for quadratic functions, but it can be generalized to any $L$-smooth and $\mu$-strongly convex functions, assuming the initial point $x^0$ is close enough to the optimal point $x_*$ (see Theorem 6 in Appendix C).

**Theorem 3.** *Let's denote $\theta = \{\alpha, \beta, \nu\}$[1]. For any function $F(x) = x^T Ax + b^T x + c$ that satisfies $0 < \mu \leq \lambda_i(A) \leq L$ for all $i = 1, \ldots, n$ and any $x^0$, $\exists\{\epsilon_k\}$, with $\epsilon_k \geq 0$, such that the deterministic QHM algorithm $z^{k+1} = Tz^k$ satisfies*

$$\left\| x^k - x_* \right\| \leq (R(\theta, \mu, L) + \epsilon_k)^k \left\| x^0 - x_* \right\|,$$

*where $x_* = \arg\min_x F(x)$, $\lim_{k\to\infty} \epsilon_k = 0$ and $R(\theta, \mu, L) = \rho(T)$, which can be characterized as*

$$R(\theta, \mu, L) = \max\{r(\theta, \mu), r(\theta, L)\}, \quad \text{where}$$

$$r(\theta, \lambda) = \begin{cases} 0.5\left(\sqrt{C_1(\lambda)^2 - 4C_2(\lambda)} + C_1(\lambda)\right) & \text{if } C_1(\lambda) \geq 0, C_1(\lambda)^2 - 4C_2(\lambda) \geq 0, \\ 0.5\left(\sqrt{C_1(\lambda)^2 - 4C_2(\lambda)} - C_1(\lambda)\right) & \text{if } C_1(\lambda) < 0, C_1(\lambda)^2 - 4C_2(\lambda) \geq 0, \\ \sqrt{C_2(\lambda)} & \text{if } C_1(\lambda)^2 - 4C_2(\lambda) < 0, \end{cases} \tag{12}$$

$$C_1(\lambda, \theta) = 1 - \alpha\lambda + \alpha\lambda\nu\beta + \beta,$$

$$C_2(\lambda, \theta) = \beta(1 - \alpha\lambda + \alpha\lambda\nu).$$

*To ensure $R(\theta, \mu, L) < 1$, the parameters $\alpha, \beta, \nu$ must satisfy the following constraints:*

$$0 < \alpha < \frac{2(1+\beta)}{L(1 + \beta(1 - 2\nu))}, \qquad 0 \leq \beta < 1, \qquad 0 \leq \nu \leq 1. \tag{13}$$

*In addition, the optimal rate depends only on $\kappa$: $\min_\theta R(\theta, \mu, L)$ is a function of only $\kappa$.*

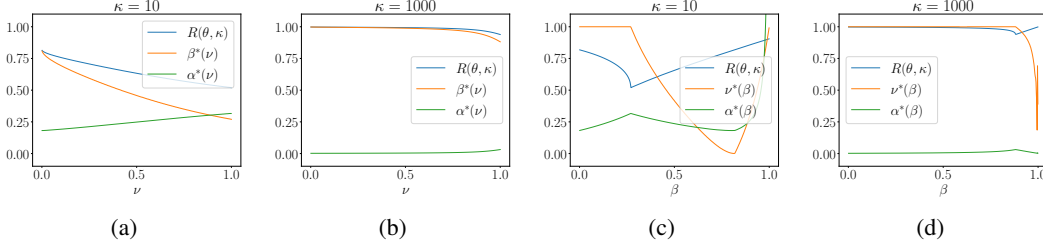

| (a) | (b) | (c) | (d) |

Figure 1: Plots (a), (b) show the dependence of the optimal $\alpha, \beta$ and convergence rate as a function of $\nu$. We can see that both rate and optimal $\beta$ are decreasing functions of $\nu$. Plots (c), (d) show the dependence of optimal $\alpha, \nu$ and rate on $\beta$. We can see that there are three phases in which the dependence is quite different. Also note that in all presented cases, changing $\nu$ required changing $\alpha$ in the same way (they increase and decrease together).

The conditions in (13) characterize the stability region of QHM. Note that when $\nu = 0$ we have the classical result for gradient descent: $\alpha < 2/L$; when $\nu = 1$, the condition matches that of the normalized heavy ball: $\alpha < 2(1 + \beta)/(L(1 - \beta))$.

The equations (12) define the convergence rate for any fixed values of the parameters $\alpha, \beta, \nu$. While it does not give a simple analytic form, it allows us to conduct easy numerical investigations. To gain more intuition into the effect that momentum parameters $\nu$ and $\beta$ have on the convergence rate, we study how the optimal $\nu$ changes as a function of $\beta$ and vice versa. To find the optimal parameters and rate, we solve the corresponding optimization problem numerically (using the procedure described in Appendix D). For each pair $\{\beta, \nu\}$ we set $\alpha$ to the optimal value in order to remove its effect. These plots are presented in Figure 1.

A natural way to think about the interplay between parameters $\alpha, \beta$ and $\nu$ is in terms of the total "amount of momentum". Intuitively, it should be controlled by the product of $\nu \times \beta$. This intuition helps explain Figure 1 (a), (b), which show the dependence of the optimal $\beta$ as a function of $\nu$ for different values of $\kappa$. We can see that for bigger values of $\nu$ we need to use smaller values of $\beta$, since increasing each one of them increases the "amount of momentum" in QHM. However, the same intuition fails when considering $\nu$ as a function of $\beta$ (and $\beta$ is big enough), as shown in Figure 1 (c), (d). In this case there are 3 regimes of different behavior. In the first regime, since $\beta$ is small, the amount of momentum is not enough for the problem and thus the optimal $\nu$ is always 1. In this phase we also need to increase $\alpha$ when increasing $\beta$ (it is typical to use larger learning rate when the momentum coefficient is bigger). The second phase begins when we reach the optimal value of $\beta$ (rate is minimal) and, after that, the amount of momentum becomes too big and we need to decrease $\nu$ and $\alpha$. However, somewhat surprisingly, there is a third phase, when $\beta$ becomes big enough we need to start increasing $\nu$ and $\alpha$ again. Thus we can see that it's not just the product of $\nu \beta$ that governs the behavior of QHM, but a more complicated function.

Finally, based on our analytic and numerical investigations, we conjecture that the optimal convergence rate is a *monotonically decreasing* function of $\nu$ (if $\alpha$ and $\beta$ are chosen optimally for each $\nu$). While we can't prove this statement[2], we verify this conjecture numerically in Appendix D. The code of all of our experiments is available at `https://github.com/Kipok/understanding-momentum`.

## 5   Stationary analysis

In this section, we study the stationary behavior of QHM with constant parameters $\alpha, \beta$ and $\nu$. Again we only consider quadratic functions for the same reasons as outlined in the beginning of Section 4. In other words, we focus on the linear dynamics of (10) driven by the noise $\xi^k$ as $k \to \infty$ (where the deterministic part depending on $x^0$ dies out). Under the assumptions of Section 4 we have the following result on the covariance matrix defined as $\Sigma_x \triangleq \lim_{k \to \infty} \mathbf{E}[x^k(x^k)^T]$.

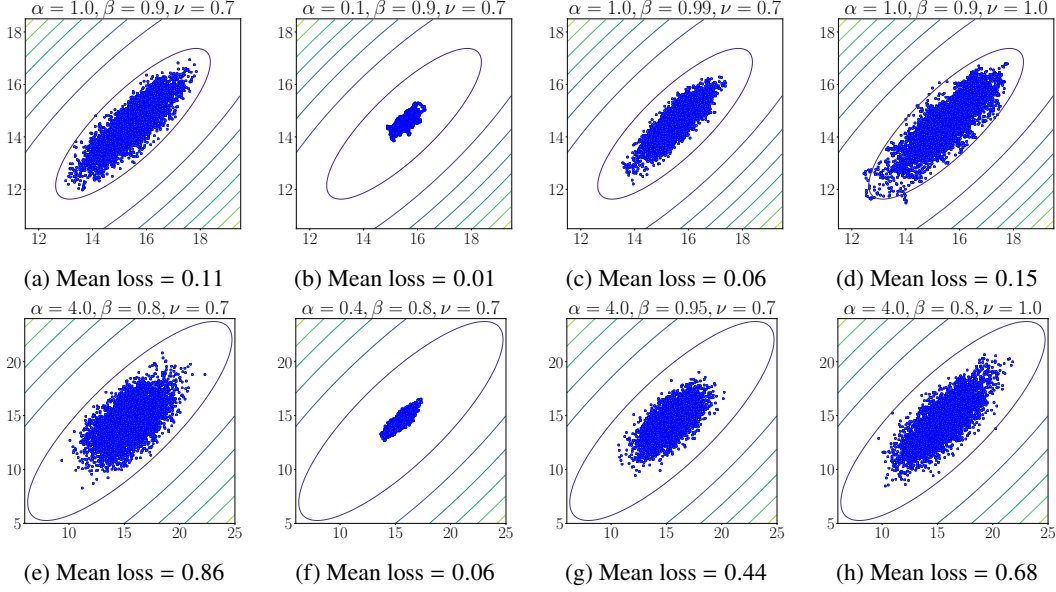

Figure 2: Changes in the shape and size of stationary distribution changes with respect to $\alpha, \beta$, and $\nu$ on a 2-dimensional quadratic problem. Each picture shows the last 5000 iterates of QHM on a contour plot. The first picture of each row is a reference and other pictures should be compared to it. The second pictures show how the stationary distribution changes when we decrease $\alpha$. The third and fourth show the dependence on $\beta$ and $\nu$, respectively. We can see that as expected, moving $\alpha \to 0$ and $\beta \to 1$ always decreases the achievable loss. However, the dependence on $\nu$ is more complicated, and for some values of $\alpha$ and $\beta$ increasing $\nu$ increases the loss (top row), while for other values the dependence is reversed (bottom row). Note the scale change between top and bottom plots.

**Theorem 4.** *Suppose $F(x) = \frac{1}{2}x^T A x$, where $A$ is symmetric positive definite matrix. The stochastic gradients satisfy $g^k = \nabla F(x^k) + \xi$, where $\xi$ is a random vector independent of $x^k$ with zero mean $\mathbf{E}[\xi] = 0$ and covariance matrix $\mathbf{E}[\xi \xi^T] = \Sigma_\xi$. Also, suppose the parameters $\alpha, \beta, \nu$ satisfy (13). Then the QHM algorithm (6), equivalently (10) in this case, converges to a stationary distribution satisfying*

$$A\Sigma_x + \Sigma_x A = \alpha A \Sigma_\xi + O(\alpha^2). \tag{14}$$

When $\nu = 1$, this result matches the known formula for the stationary distribution of unnormalized SHB [19] with reparametrization of $\alpha \to \alpha/(1 - \beta)$. Note that Theorem 4 shows that for the normalized version of the algorithm, the stationary distribution's covariance does not depend on $\beta$ (or $\nu$) to the first order in $\alpha$. In order to explore such dependence, we need to expand the dependence on $\alpha$ to the second order. In that case, we are not able to obtain a matrix equation, but can get the following relation for $\mathbf{tr}(A\Sigma_x)$.

**Theorem 5.** *Under the conditions of Theorem 4, we have*

$$\mathbf{tr}(A\Sigma_x) = \frac{\alpha}{2}\mathbf{tr}(\Sigma_\xi) + \frac{\alpha^2}{4}\left(1 + \frac{2\nu\beta}{1-\beta}\left[\frac{2\nu\beta}{1+\beta} - 1\right]\right)\mathbf{tr}(A\Sigma_\xi) + O(\alpha^3). \tag{15}$$

We note that $\mathbf{tr}(A\Sigma_x)$ is twice the mean value of $F(x)$ when the dynamics have reached stationarity, so the right-hand side of (15) is approximately the "achievable loss" given the values of $\alpha, \beta$ and $\nu$. It is interesting to consider several special cases:

- $\nu = 0$ (SGD): $\mathbf{tr}(A\Sigma_x) = \frac{\alpha}{2}\mathbf{tr}(\Sigma_\xi) + \frac{\alpha^2}{4}\mathbf{tr}(A\Sigma_\xi) + O(\alpha^3)$.

- $\nu = 1$ (SHB): $\mathbf{tr}(A\Sigma_x) = \frac{\alpha}{2}\mathbf{tr}(\Sigma_\xi) + \frac{\alpha^2}{4}\frac{1-\beta}{1+\beta}\mathbf{tr}(A\Sigma_\xi) + O(\alpha^3)$.

- $\nu = \beta$ (NAG): $\mathbf{tr}(A\Sigma_x) = \frac{\alpha}{2}\mathbf{tr}(\Sigma_\xi) + \frac{\alpha^2}{4}\left(1 - \frac{2\beta^2(1+2\beta)}{1+\beta}\right)\mathbf{tr}(A\Sigma_\xi) + O(\alpha^3)$.

From the expressions for SHB and NAG, it might be beneficial to move $\beta$ to 1 during training in order to make the achievable loss smaller. While moving $\beta$ to 1 is somewhat counter-intuitive, we

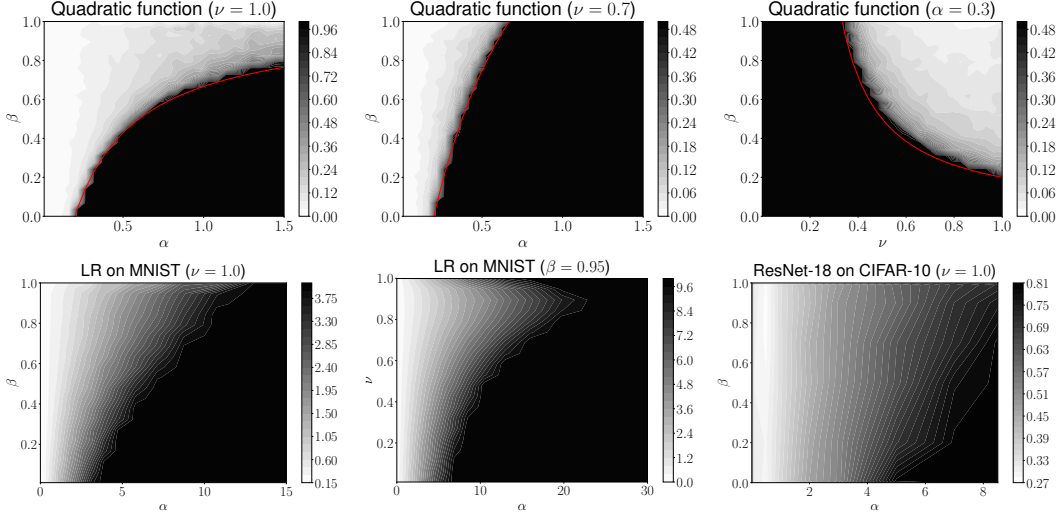

Figure 3: These pictures show dependence of the average final loss (depicted with color: whiter is smaller) on the parameters of QHM algorithm for different problems. The top row shows results for a synthetic 2-dimensional quadratic problem, where all the assumptions of Theorem 5 are satisfied. The red curve indicates the boundary of convergence region (algorithm diverges below it). In this case, we start the algorithm directly at the optimal value to measure the size of stationary distribution and ignore convergence rate. We can see that as predicted by theory, smaller $\alpha$ and bigger $\beta$ make the final loss smaller. The bottom row shows results of the same experiments repeated for logistic regression on MNIST and ResNet-18 on the CIFAR-10 dataset. We can see that while the assumptions of Theorem 5 are no longer valid, QHM still shows similar qualitative behavior.

proved in Section 3 that QHM still converges asymptotically in this regime, assuming $\nu$ also goes to 1 and $\nu\beta$ converges to 1 "slower" than $\alpha$ converges to 0. However, since we only consider Taylor expansion in $\alpha$, there is no guarantee that the approximation remains accurate when $\nu$ and $\beta$ converge to 1 (see Appendix G for evaluation of this approximation error). In order to precisely investigate the dependence on $\beta$ and $\nu$, it is necessary to further extend our results by considering Taylor expansion with respect to them as well, especially in terms of $1 - \beta$. We leave this for future work.

Figure 2 shows a visualization of the QHM stationary distribution on a 2-dimensional quadratic problem. We can see that our prediction about the dependence on $\alpha$ and $\beta$ holds in this case. However, the dependence on $\nu$ is more complicated: the top and bottom rows of Figure 2 show opposite behavior. Comparing this experiment with our analysis of the convergence rate (Figure 1) we can see another confirmation that for big values of $\beta$, increasing $\nu$ can, in a sense, decrease the "amount of momentum" in the system. Next, we evaluate the average final loss for a large grid of parameters $\alpha, \beta$ and $\nu$ on three problems: a 2-dimensional quadratic function (where all of our assumptions are satisfied), logistic regression on the MNIST [16] dataset (where the quadratic assumption is approximately satisfied, but gradient noise comes from mini-batches) and ResNet-18 [10] on CIFAR-10 [13] (where all of our assumptions are likely violated). Figure 3 shows the results of this experiment. We can indeed see that $\beta \to 1$ and $\alpha \to 0$ make the final loss smaller in all cases. The dependence on $\nu$ is less clear, but we can see that for large values of $\beta$ it is approximately quadratic, with a minimum at some $\nu < 1$. Thus from this point of view $\nu \neq 1$ helps when $\beta$ is big enough, which might be one of the reasons for the empirical success of the QHM algorithm. Notice that the empirical dependence on $\nu$ is qualitatively the same as predicted by formula (15), but with optimal value shifted closer to 1. See Appendix F for details.

## 6 Some practical implications and guidelines

In this section, we present some practical implications and guidelines for setting learning rate and momentum parameters in practical machine learning applications. In particular, we consider the question of how to set the optimal parameters in each stage of the popular constant-and-drop scheme for deep learning. We argue that in order to answer this question, it is necessary to consider both

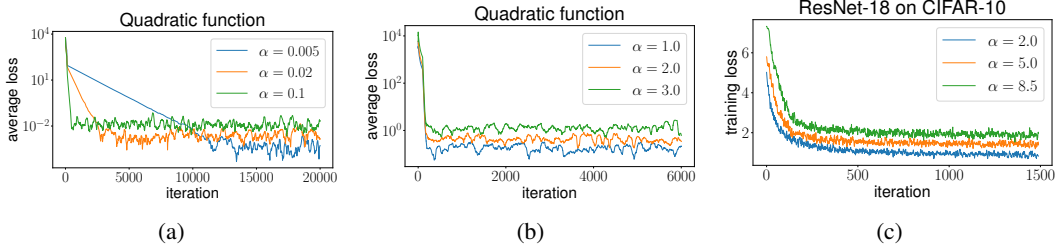

Figure 4: (a) This plot shows a trade-off between stationary distribution size (final loss) and convergence rate on a simple 2-dimensional quadratic problem. Algorithms that converge faster, typically will converge to a higher final loss. (b) This plot illustrates the regime where there is no trade-off between stationary distribution size and convergence rate. Larger values of $\alpha$ don't change the convergence rate, while making final loss significantly higher. To make plots (a) and (b) smoother, we plot the average value of the loss for each 100 iterations on $y$-axis. (c) This plot shows that the same behavior can also be observed in training deep neural networks. For all plots $\beta = 0.9, \nu = 1.0$. All of the presented results depend continuously on the algorithm's parameters (e.g. the transition between behaviours shown in (a) and (b) is smooth).

convergence rate and stationary distribution perspectives. There is typically a trade-off between obtaining a fast rate and a small stationary distribution. You can see an illustration of this trade-off in Figure 4 (a). Interestingly, by combining stationary analysis of Section 5 and results for the convergence rate (3), we can find certain regimes of parameters $\alpha, \beta$, and $\nu$ where the final loss and the convergence speed do not compete with each other.

One of the most important of these regimes happens in the case of the SHB algorithm ($\nu = 1$). In that case, we can see that when $C_1^2(l) - C_2^2(l) \le 0, l \in \{\mu, L\}$, the convergence rate equals $\sqrt{\beta}$ and does not depend on $\alpha$. Thus, as long as this inequality is satisfied, we can set $\alpha$ as small as possible and it would not harm the convergence rate, but will decrease the size of stationary distribution. To get the best possible convergence rate, we, in fact, have to set $\alpha$ and $\beta$ in such a way that this inequality will turn into equality and thus there will be only a single value of $\alpha$ that could be used. However, as long as $\beta$ is not exactly at the optimal value, there is going to be some freedom in choosing $\alpha$ and it should be used to decrease the size of stationary distribution. From this point of view, the optimal value of $\alpha = \left(1 - \sqrt{\beta}\right) / \left(\mu\left(1 + \sqrt{\beta}\right)\right)$, which will be smaller then the largest possible $\alpha$ for convergence as long as $\kappa > 2$ and $\beta$ is set close to 1 (see proof of Theorem 3 for more details). This guideline contradicts some typical advice to set $\alpha$ as big as possible while algorithm still converges[3]. The refined guideline for the constant-and-drop scheme would be to set $\alpha$ as small as possible until the convergence noticeably slows down. You can see an illustration of this behavior on a simple quadratic problem (Figure 4 (b)), as well as for ResNet-18 on CIFAR-10 (Figure 4 (c)). Such regimes of no trade-off can be identified for $\beta$ and $\nu$ as well.

## 7  Conclusion

Using the general formulation of QHM, we have derived a unified set of new analytic results that give us better understanding of the role of momentum in stochastic gradient methods. Our results cover several different aspects: asymptotic convergence, stability region and local convergence rate, and characterizations of stationary distribution. We show that it is important to consider these different aspects together to understand the key trade-offs in tuning the learning rate and momentum parameters for better performance in practice. On the other hand, we note that the obtained guidelines are mainly for stochastic optimization, meaning the minimization of the *training* loss. There is evidence that different heuristics and guidelines may be necessary for achieving better generalization performance in machine learning, but this topic is beyond the scope of our current paper.

## Footnotes

[1]We drop the dependence of some functions on $\theta$ for brevity.

[2]In fact, we hypothesise that $R^*(\nu, \kappa)$ might not have analytical formula, since it is possible to show that the optimization problem over $\alpha$ and $\beta$ is equivalent to the system of highly non-linear equations.

[3]For example [8] says "So set $\beta$ as close to 1 as you can, and then find the highest $\alpha$ which still converges. Being at the knife's edge of divergence, like in gradient descent, is a good place to be."

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
