[Supplementary Material · um-apdx-final.pdf]

# Appendix

## A    From NAG to QHM

In this appendix we will mention exact steps needed to come from the original NAG formulation to the formulation assumed by the QHM algorithm. We refer the reader to the ([34] Appendix A.1) for the derivation of NAG as the following momentum method [4]

$$d_k = \beta_{k-1} d_{k-1} - \alpha_{k-1} \nabla f(x_{k-1} + \beta_{k-1} d_{k-1})$$
$$x_k = x_{k-1} + d_k$$

Next, we will move the learning rate out of the momentum into the iterates update:

$$d_k = \beta_{k-1} d_{k-1} + \nabla f(x_{k-1} - \alpha_{k-1} \beta_{k-1} d_{k-1})$$
$$x_k = x_{k-1} - \alpha_{k-1} d_k$$

When $\alpha_k$ and $\beta_k$ are constant, the two methods produce the same sequence of iterates $x_k$ if $d_0$ is initialized at 0. To make the notation more similar to the QHM algorithm, let's move all indices (except for $d_k$) up by 1:

$$d_k = \beta_k d_{k-1} + \nabla f(x_k - \alpha_k \beta_k d_{k-1})$$
$$x_{k+1} = x_k - \alpha_k d_k$$

This again does not change the algorithm. Now, let's normalize the momentum update by $1 - \beta_k$:

$$d_k = \beta_k d_{k-1} + (1 - \beta_k) \nabla f(x_k - \alpha_k \beta_k d_{k-1})$$
$$x_{k+1} = x_k - \alpha_k d_k$$

This version is equivalent to the unnormalized by re-scaling $\alpha \rightarrow \alpha/(1 - \beta)$ for constant parameters[5]. Finally, following [2] we need to make a change of variables $y_k = x_k - \alpha_k \beta_k d_{k-1}$ and additionally assume that $\beta_k = \beta$ is constant:

$$d_k = \beta d_{k-1} + (1 - \beta) \nabla f(y_k)$$
$$y_{k+1} = x_{k+1} - \alpha_k \beta d_k = x_k - \alpha_k d_k - \alpha_k \beta d_k = y_k + \alpha_k \beta d_{k-1} - \alpha_k d_k - \alpha_k \beta d_k$$
$$= y_k + \alpha_k (d_k - (1 - \beta) \nabla f(y_k)) - \alpha_k d_k - \alpha_k \beta d_k$$
$$= y_k - \alpha_k [(1 - \beta) \nabla f(y_k) + \beta d_k]$$

Renaming $y_k$ back to $x_k$ and replacing $\nabla f(y_k)$ with stochastic gradient if necessary we obtain the exact formula used in QHM update.

Overall, assuming $d_0 = 0$ and $\beta_k$ is constant, the QHM version of NAG is indeed equivalent (up to a change of variable) to the original NAG with re-scaling of $\alpha \rightarrow \alpha/(1 - \beta)$. However, if $\beta_k$ is changing from iteration to iteration, the two algorithms are no longer equivalent.

## B    Asymptotic Convergence Proofs

In this section we prove Theorems 1 and 2. For simplicity, we assume throughout that $\alpha_k$, $\nu_k$ and $\beta_k$ are nonrandom.

*Proof of Theorem 1.* Here we generalize the meta-analysis of Ruszczyński and Syski [33] to include $\nu_k$.

Summing over $k$ from $k_1$ to $\infty$ and using Assumption A.2, we get that:

$$\frac{\epsilon}{2} \sum_{k=k_1}^{\infty} \alpha_k (1 - \nu_k \beta_k) ||\nabla F(x^k)|| \leq F(x^{k_1}) - F^* + \sum_{k=k_1}^{\infty} W_k.$$

The right-hand-side is finite by (27). But since $||\nabla F(x^k)|| \geq \epsilon$ for all $k \geq k_1$ and $\beta_k \leq \bar{\beta} < 1$ and $0 \leq \nu_k \leq 1$, we have:

$$\frac{\epsilon^2}{2}(1 - \bar{\beta}) \sum_{k=k_1}^{\infty} \alpha_k \leq F(x^{k_1}) - F^* + \sum_{k=k_1}^{\infty} W_k.$$

This implies $\sum_{k=k_1}^{\infty} \alpha_k < \infty$, which contradicts (20). So we must have (7), i.e. $\liminf_k ||\nabla F(x^k)|| = 0$.

To prove (8), we consider two cases. First, assume there exists $k_0$ such that $||\nabla F(x^k)|| \geq S_k$ for all $k \geq k_0$. Then by (7), there exists a subsequence $\mathcal{K} \subset \mathbb{N}$ such that

$$\lim_{k \in \mathcal{K}, k \to \infty} ||\nabla F(x^k)|| = 0.$$

For every $l$, define the index $k(l) = \max\{k \in \mathcal{K} : k < l\}$. Since $\mathcal{K}$ is infinite, $k(l) \to \infty$ as $l \to \infty$. Then for sufficiently large $l$, i.e., when $k(l) \geq k_1$, (25) becomes

$$F(x^l) \leq F(x^{k(l)}) + \sum_{i=k(l)}^{l-1} W_i.$$

As $l \to \infty$, because of (27) and $k(l) \to \infty$, we get $\sum_{i=k(l)}^{l-1} W_i \to 0$, so

$$\limsup_{l \to \infty} F(x^l) \leq \limsup_{l \to \infty} F(x^{k}(l)) \leq \limsup_{k \in \mathcal{K}, k \to \infty} F(x^k). \tag{30}$$

Since the reverse inequality is trivial, we obtain (8).

In the second case, we have $||\nabla F(x^k)|| < S_k$ fulfilled infinitely often. In that case, for each $l$ define the index $k(l) = \max\{k : k < l \text{ and } ||\nabla F(x^k)|| < S_k\}$. As before, $k(l) \to \infty$ as $l \to \infty$. Furthermore, (26) implies $||\nabla F(x^{k(l)})|| \to 0$ as $l \to \infty$. Therefore, there exists $\mathcal{K} \subset \mathbb{N}$ with $\{k(l)\}_l \subset \mathcal{K}$ and $\lim_{k \in \mathcal{K}, k \to \infty} ||\nabla F(x^k)|| = 0$. In this case, we obtain from (25) that

$$F(x^l) \leq F(x^{k(l)}) + \alpha_{k(l)}(1 - \nu_{k(l)}\beta_{k(l)})||\nabla F(x^{k(l)})||S_{k(l)} + \sum_{i=k(l)}^{l-1} W_i.$$

Because $\alpha_k < \bar{\alpha} < \infty$ for all $k$, $\nu_k$, $\beta_k$ are in $[0, 1]$, and $||\nabla F(x^{k(l)})|| \to 0$, the latter two terms in the above inequality converge to zero as $l \to \infty$. So we obtain (30) again. This concludes the proof of the lemma. □

All that remains now is to use the smoothness inequality (24) to identify the sequences $S_k$ and $W_k$ for the dynamics of the modified algorithm (16)-(18), and prove (26) and (27).

From the update formula (18) and using $g^k = \nabla F(x^k) + \xi^k$, we obtain

$$\begin{aligned}
\Delta x^{k+1} = x^{k+1} - x^k &= -\alpha_k b^k \\
&= -\alpha_k b^k + \alpha_k(1 - \nu_k \beta_k)(g^k - \nabla F(x^k) - \xi^k) \\
&= -\alpha_k(1 - \nu_k \beta_k)\nabla F(x^k) - \alpha_k(b^k - (1 - \nu_k \beta_k)g^k) - \alpha_k(1 - \nu_k \beta_k)\xi^k.
\end{aligned}$$

By the smoothness assumption A.1, we have

$$F(x^{k+1}) \le F(x^k) + \langle \nabla F(x^k), \Delta x^{k+1} \rangle + \frac{L}{2} \left\| \Delta x^{k+1} \right\|^2$$

$$= F(x^k) - \alpha_k (1 - \nu_k \beta_k) \left\| \nabla F(x^k) \right\|^2 - \alpha_k \langle \nabla F(x^k), b^k - (1 - \nu_k \beta_k) g^k \rangle$$

$$- \alpha_k (1 - \nu_k \beta_k) \langle \nabla F(x^k), \xi^k \rangle + \frac{L}{2} \left\| \Delta x^{k+1} \right\|^2$$

$$\le F(x^k) - \alpha_k (1 - \nu_k \beta_k) \left\| \nabla F(x^k) \right\|^2 + \alpha_k \left\| \nabla F(x^k) \right\| \cdot \left\| b^k - (1 - \nu_k \beta_k) g^k \right\|$$

$$- \alpha_k (1 - \nu_k \beta_k) \langle \nabla F(x^k), \xi^k \rangle + \frac{L}{2} \left\| \Delta x^{k+1} \right\|^2$$

$$= F(x^k) - \alpha_k (1 - \nu_k \beta_k) \left\| \nabla F(x^k) \right\| \left( \left\| \nabla F(x^k) \right\| - \frac{\left\| b^k - (1 - \nu_k \beta_k) g^k \right\|}{(1 - \nu_k \beta_k)} \right)$$

$$- \alpha_k (1 - \nu_k \beta_k) \langle \nabla F(x^k), \xi^k \rangle + \frac{L}{2} \left\| \Delta x^{k+1} \right\|^2$$

Comparing with (25), we define

$$S_k = \frac{\left\| b^k - (1 - \nu_k \beta_k) g^k \right\|}{1 - \nu_k \beta_k}, \tag{31}$$

$$W_k = -\alpha_k (1 - \nu_k \beta_k) \langle \nabla F(x^k), \xi^k \rangle + \frac{L}{2} \left\| \Delta x^{k+1} \right\|^2. \tag{32}$$

First we show $S_k \to 0$. From the update formula, we have

$$b^k = (1 - \nu_k) g^k + \nu_k \left( (1 - \beta_k) g^k + i_k \beta_k d^{k-1} \right) = (1 - \nu_k \beta_k) g^k + i_k \nu_k \beta_k d^{k-1}.$$

Then because $i_k \|d^{k-1}\| \le \rho$, $\beta_k \to 0$, and $\sup_k \beta_k = \bar{\beta} < 1$, we have

$$\lim_{k \to \infty} S_k = \lim_{k \to \infty} \frac{\|b^k - (1 - \nu_k \beta_k) g^k\|}{1 - \nu_k \beta_k} = \lim_{k \to \infty} \frac{i_k \nu_k \beta_k \|d^{k-1}\|}{1 - \nu_k \beta_k} \le \lim_{k \to \infty} \frac{\nu_k \beta_k \rho}{1 - \nu_k \bar{\beta}} \le \lim_{k \to \infty} \frac{\beta_k \rho}{1 - \bar{\beta}} = 0.$$

Because $S_k \ge 0$, $\lim_k S_k = 0$.

Now we show that $W_k$ is summable almost surely. To begin, we need to show that $\|\Delta x^{k+1}\|^2$ is summable, for which we need the following lemma:

**Lemma 2.** *There is a random variable $C$, constant in $k$, such that $\mathbf{E}_k[\|b^k\|^2] \le C$ for all $k$ almost surely.*

*Proof.* To see this, observe that

$$\|b^k\| = \|(1 - \nu_k) g^k + \nu_k \left( (1 - \beta_k) g^k + i_k \beta_k d^{k-1} \right)\| = \|g^k - \nu_k g^k + \nu_k g^k - \nu_k \beta_k g^k + i_k \nu_k \beta_k d^{k-1}\|$$

$$= \|(1 - \nu_k \beta_k) g^k + i_k \nu_k \beta_k d^{k-1}\| \le (1 - \nu_k \beta_k) \|g^k\| + i_k \nu_k \beta_k \|d^{k-1}\| \le (1 - \nu_k \beta_k) \|g^k\| + \nu_k \beta_k \rho,$$

where in the last inequality we used $i_k \|d^{k-1}\| \le \rho$. Then

$$\mathbf{E}_k[\|b^k\|^2] \le (1 - \nu_k \beta_k)^2 \mathbf{E}_k[\|g^k\|^2] + (1 - \nu_k \beta_k) \nu_k \beta_k \rho \mathbf{E}_k[\|g^k\|] + \nu_k^2 \beta_k^2 \rho^2$$

By assumption A.2 and A.3, the first and second conditional moments $\mathbf{E}_k[\|g^k\|]$ and $\mathbf{E}_k[\|g^k\|^2]$ are both bounded uniformly in $k$. Then because $\nu_k$ and $\beta_k$ are in $[0, 1]$ and $\rho$ is constant in $k$, we can put

$$\mathbf{E}_k[\|b^k\|^2] \le (1 - \nu_k \beta_k)^2 C' + (1 - \nu_k \beta_k) \nu_k \beta_k \rho C'' + \nu_k^2 \beta_k^2 \rho^2 \le C,$$

which is what we wanted. Note that $C$ could be a random variable (depending on $\omega$), but this bound holds almost surely. $\qquad \square$

**Lemma 3.** $\sum_{k=1}^{\infty} \|\Delta x^{k+1}\|^2 < \infty$ almost surely.

*Proof.* We will use the following useful proposition (known as Levy's sharpening of Borel-Cantelli Lemma, see e.g. Meyer [20, Chapter 1, Theorem 21]):

**Proposition 1.** *Let $\{b_k\}$ be a sequence of positive, integrable random variables, and let $a_i = \mathbf{E}[b_i|\mathcal{F}_i]$, where $\mathcal{F}_i = \{b_0, \dots b_{i-1}\}$. Then defining the partial sums $B_k = \sum_{i=1}^k b_i$, $A_k = \sum_{i=1}^k a_i$,*

$$\lim_{k\to\infty} A_k < \infty \text{ a.s.} \implies \lim_{k\to\infty} B_k < \infty \text{ a.s.}$$

So to prove $\sum_{k=1}^\infty ||\Delta x^{k+1}||^2 < \infty$, we only need to prove $\sum_{k=1}^\infty \mathbf{E}_k[||\Delta x^{k+1}||^2] < \infty$. To see this, observe that

$$\sum_{k=1}^\infty \mathbf{E}_k[||\Delta x^{k+1}||^2] = \sum_{k=1}^\infty \alpha_k^2 \mathbf{E}_k[||b^k||^2],$$

so because $\mathbf{E}_k[||b^k||^2] \le C$ a.s.,

$$\sum_{k=1}^\infty \mathbf{E}_k[||\Delta x^{k+1}||^2] \le C \sum_{k=1}^\infty \alpha_k^2 < \infty \text{ a.s.},$$

where we used (21). Applying the proposition finishes the lemma. $\qquad\square$

The last term remaining in $W_k$ is $-\alpha_k(1 - \nu_k\beta_k)\langle\nabla F(x^k), \xi^k\rangle$. We show that

$$M_k = \sum_{i=0}^k \alpha_i(1 - \nu_i\beta_i)\langle\nabla F(x^i), \xi^i\rangle$$

is a convergent martingale. First, note that $\mathbf{E}_i[\alpha_i(1 - \nu_i\beta_i)\langle\nabla F(x^i), \xi^i\rangle] = 0$, so $E_k[M_k] = M_{k-1}$, and $M_k$ is a martingale. Now we show that $\sup_k \mathbf{E}[M_k^2]$ is bounded, which will imply a.s. convergence by Doob's forward convergence theorem [38, Section 11.5]. Indeed, $\mathbf{E}[M_k^2] = \mathbf{E}[M_0^2] + \sum_{i=1}^k \mathbf{E}[(M_i - M_{i-1})^2]$ [38, Section 12.1].

$$\mathbf{E}[M_0^2] = \mathbf{E}[\langle\nabla F(x^0), \alpha_0(1-\nu_0\beta_0)\xi^0\rangle^2] \le \mathbf{E}[||\nabla F(x^0)||^2 \cdot \alpha_0^2(1-\nu_0\beta_0)^2||\xi^0||^2] \le G\mathbf{E}[\alpha_0^2(1-\nu_0\beta_0)^2||\xi^0||^2].$$

Because $\xi^0$ depends only on $x_0$, we can upper bound this expectation with some constant $C$ by using Assumption A.3. Then we have that $\mathbf{E}[M_0^2] \le C$, so $\mathbf{E}[M_k^2] \le C + \sum_{i=1}^k \mathbf{E}[(M_i - M_{i-1})^2]$. Therefore,

$$\sup_k \mathbf{E}[M_k^2] \le C + \sup_k \sum_{i=1}^k \mathbf{E}[(\alpha_i(1 - \nu_i\beta_i)\langle\nabla F(x^i), \xi^i\rangle)^2] \le C + G^2 \sum_{i=1}^\infty \mathbf{E}[\alpha_i^2||\xi^i||^2],$$

where the last inquality used Assumption A.2 and the fact that $0 \le \nu_i\beta_i \le 1$ almost surely. Moreover,

$$\sum_{i=1}^\infty \mathbf{E}[\alpha_i^2||\xi^i||^2] = \sum_{i=1}^\infty \mathbf{E}[\mathbf{E}_i[\alpha_i^2||\xi^i||^2]] = \sum_{i=1}^\infty \mathbf{E}[\alpha_i^2\mathbf{E}_i[||\xi^i||^2]] \le C_2 \sum_{i=1}^\infty \mathbf{E}[\alpha_i^2],$$

where the first equality is by the law of total expectation and the inequality comes from Assumption A.3. Because $\sum_{i=1}^\infty \alpha_i^2 < \infty$, we finally have

$$\sup_k \mathbf{E}[M_k^2] < \infty,$$

so $M_k$ is a convergent martingale. In particular,

$$\sum_{k=0}^\infty -\alpha_k(1 - \nu_k\beta_k)\langle\nabla F(x^k), \xi^k\rangle > -\infty \text{ a.s.}$$

Combining this with Lemma 3, we get

$$\sum_{k=0}^\infty W_k < \infty \text{ a.s..}$$

We have shown (26) and (27), which concludes the proof. $\qquad\square$

Now we prove Theorem 2, where under a stronger noise assumption we show that $\beta_k \to 1$ is admissible as long as it goes to 1 slow enough.

*Proof of Theorem 2.* Assume the sequences $\{\alpha_k\}$, $\{\beta_k\}$, and $\{\nu_k\}$ satisfy the following:

$$\sum_{k=0}^{\infty} \alpha_k = \infty \tag{33}$$

$$\sum_{k=0}^{\infty} (1 - \nu_k \beta_k)^2 < \infty \tag{34}$$

$$\sum_{k=0}^{\infty} \frac{\alpha_k^2}{1 - \nu_k \beta_k} < \infty \tag{35}$$

$$\lim_{k \to \infty} \beta_k = 1 \tag{36}$$

then sequence $\{x^k\}$ generated by the algorithm (6) satisfies

$$\liminf_{k \to \infty} \|\nabla F(x^k)\| = 0 \quad a.s. \tag{37}$$

By the smoothness assumption, we have

$$
\begin{aligned}
F(x^{k+1}) &\leq F(x^k) + \langle \nabla F(x^k), x^{k+1} - x^k \rangle + \frac{L}{2}\|x^{k+1} - x^k\|^2 \\
&= F(x^k) + \langle \nabla F(x^k), -\alpha_k b^k \rangle + \frac{L}{2}\alpha_k^2 \|b^k\|^2 \\
&= F(x^k) + \left\langle \nabla F(x^k), -\alpha_k \left( \nabla F(x^k) + b^k - \nabla F(x^k) \right) \right\rangle + \frac{L}{2}\alpha_k^2 \|b^k\|^2 \\
&= F(x^k) - \alpha_k \|\nabla F(x^k)\|^2 - \alpha_k \langle \nabla F(x^k), b^k - \nabla F(x^k) \rangle + \frac{L}{2}\alpha_k^2 \|b^k\|^2. \tag{38}
\end{aligned}
$$

Using the update formula in (6), we have

$$
\begin{aligned}
b^k - \nabla F(x^k) &= (1 - \nu_k)g^k + \nu_k d^k - \nabla F(x^k) \\
&= (1 - \nu_k \beta_k)g^k + \nu_k \beta_k d^{k-1} - \nabla F(x^k) \\
&= (1 - \nu_k \beta_k)(g^k - \nabla F(x^k)) + \nu_k \beta_k (d^{k-1} - \nabla F(x^k)) \\
&= (1 - \nu_k \beta_k)\xi^k + \nu_k \beta_k (d^{k-1} - \nabla F(x^k)). \tag{39}
\end{aligned}
$$

Substitution of (39) into (38) yields

$$
\begin{aligned}
F(x^{k+1}) &\leq F(x^k) - \alpha_k \|\nabla F(x^k)\|^2 - \alpha_k \nu_k \beta_k \langle \nabla F(x^k), d^{k-1} - \nabla F(x^k) \rangle \\
&\quad - \alpha_k(1 - \nu_k \beta_k)\langle \nabla F(x^k), \xi^k \rangle + \frac{L}{2}\alpha_k^2 \|b^k\|^2 \\
&\leq F(x^k) - \alpha_k \|\nabla F(x^k)\|^2 + \alpha_k \nu_k \beta_k \left( \frac{1}{4}\|F(x^k)\|^2 + \|d^{k-1} - \nabla F(x^k)\|^2 \right) \\
&\quad - \alpha_k(1 - \nu_k \beta_k)\langle \nabla F(x^k), \xi^k \rangle + \frac{L}{2}\alpha_k^2 \|b^k\|^2 \\
&\leq F(x^k) - \frac{3\alpha_k}{4}\|\nabla F(x^k)\|^2 + \alpha_k \nu_k \beta_k \|d^{k-1} - \nabla F(x^k)\|^2 \\
&\quad - \alpha_k(1 - \nu_k \beta_k)\langle \nabla F(x^k), \xi^k \rangle + \frac{L}{2}\alpha_k^2 \|b^k\|^2. \tag{40}
\end{aligned}
$$

where in the second inequality we used $\langle a, b \rangle \leq \frac{1}{4}\|a\|^2 + \|b\|^2$ for any two vectors $a$ and $b$, and in the last inequality we used $0 \leq \nu_k \beta_k \leq 1$. Taking conditional expectation on both sides of the above inequality and using $\mathbf{E}[\xi^k] = 0$, we get

$$\mathbf{E}_k \left[ F(x^{k+1}) \right] \leq F(x^k) - \frac{3\alpha_k}{4}\|\nabla F(x^k)\|^2 + \alpha_k \nu_k \beta_k \|d^{k-1} - \nabla F(x^k)\|^2 + \frac{L}{2}\alpha_k^2 \mathbf{E}_k \left[ \|b^k\|^2 \right]. \tag{41}$$

Next we analyze the sequence $\{d^{k-1} - \nabla F(x^k)\}$. From the update formula in (6), we have

$$
\begin{aligned}
d^k - \nabla F(x^{k+1}) &= \nu_k \beta_k d^{k-1} + (1 - \nu_k \beta_k)g^k - \nabla F(x^{k+1}) + \nabla F(x^k) - \nabla F(x^k) \\
&= \nu_k \beta_k (d^{k-1} - \nabla F(x^k)) + (1 - \nu_k \beta_k)(g^k - \nabla F(x^k)) + (\nabla F(x^k) - \nabla F(x^{k+1})) \\
&= \nu_k \beta_k (d^{k-1} - \nabla F(x^k)) + (1 - \nu_k \beta_k)\xi^k + (\nabla F(x^k) - \nabla F(x^{k+1})).
\end{aligned}
$$

Therefore,

$$
\begin{aligned}
\left\|d^k - \nabla F(x^{k+1})\right\|^2 &= (\nu_k\beta_k)^2\left\|d^{k-1} - \nabla F(x^k)\right\|^2 + \left\|(1 - \nu_k\beta_k)\xi^k + \left(\nabla F(x^k) - \nabla F(x^{k+1})\right)\right\|^2 \\
&\quad + 2\nu_k\beta_k\left\langle d^{k-1} - \nabla F(x^k),\, (1 - \nu_k\beta_k)\xi^k + \left(\nabla F(x^k) - \nabla F(x^{k+1})\right)\right\rangle \\
&\leq (\nu_k\beta_k)^2\left\|d^{k-1} - \nabla F(x^k)\right\|^2 + 2(1 - \nu_k\beta_k)^2\left\|\xi^k\right\|^2 + 2\left\|\nabla F(x^k) - \nabla F(x^{k+1})\right\|^2 \\
&\quad + 2\nu_k\beta_k\left\langle d^{k-1} - \nabla F(x^k),\, (1 - \nu_k\beta_k)\xi^k\right\rangle \\
&\quad + 2\nu_k\beta_k\left\langle d^{k-1} - \nabla F(x^k),\, \left(\nabla F(x^k) - \nabla F(x^{k+1})\right)\right\rangle \\
&\leq (\nu_k\beta_k)^2\left\|d^{k-1} - \nabla F(x^k)\right\|^2 + 2(1 - \nu_k\beta_k)^2\left\|\xi^k\right\|^2 + 2\left\|\nabla F(x^k) - \nabla F(x^{k+1})\right\|^2 \\
&\quad + 2\nu_k\beta_k\left\langle d^{k-1} - \nabla F(x^k),\, (1 - \nu_k\beta_k)\xi^k\right\rangle \\
&\quad + \nu_k\beta_k\left(\eta\left\|d^{k-1} - \nabla F(x^k)\right\|^2 + \frac{1}{\eta}\left\|\nabla F(x^k) - \nabla F(x^{k+1})\right\|^2\right),
\end{aligned}
$$

where in the first inequality we used $\|a + b\|^2 \leq 2\|a\|^2 + 2\|b\|^2$, and in the second inequality we used $2\langle a, b\rangle \leq \eta\|a\|^2 + \frac{1}{\eta}\|b\|^2$ for any $\eta > 0$. By the smoothness assumption, we have

$$
\left\|\nabla F(x^k) - \nabla F(x^{k+1})\right\|^2 \leq L^2\left\|x^k - x^{k+1}\right\|^2 = \alpha_k^2 L^2\left\|b^k\right\|^2,
$$

which, combining with the previous inequality, leads to

$$
\begin{aligned}
\left\|d^k - \nabla F(x^{k+1})\right\|^2 &\leq \left((\nu_k\beta_k)^2 + \nu_k\beta_k\eta\right)\left\|d^{k-1} - \nabla F(x^k)\right\|^2 + 2(1 - \nu_k\beta_k)^2\left\|\xi^k\right\|^2 + 2\alpha_k^2 L^2\left\|b^k\right\|^2 \\
&\quad + 2\nu_k\beta_k\left\langle d^{k-1} - \nabla F(x^k),\, (1 - \nu_k\beta_k)\xi^k\right\rangle + \frac{\nu_k\beta_k}{\eta}\alpha_k^2 L^2\left\|b^k\right\|^2 \\
&\leq \nu_k\beta_k(\nu_k\beta_k + \eta)\left\|d^{k-1} - \nabla F(x^k)\right\|^2 + 2(1 - \nu_k\beta_k)^2\left\|\xi^k\right\|^2 + 2\alpha_k^2 L^2\left\|b^k\right\|^2 \\
&\quad + 2\nu_k\beta_k\left\langle d^{k-1} - \nabla F(x^k),\, (1 - \nu_k\beta_k)\xi^k\right\rangle + \frac{1}{\eta}\alpha_k^2 L^2\left\|b^k\right\|^2.
\end{aligned}
$$

Choosing $\eta = 1 - \nu_k\beta_k$, we obtain

$$
\begin{aligned}
\left\|d^k - \nabla F(x^{k+1})\right\|^2 &\leq \nu_k\beta_k\left\|d^{k-1} - \nabla F(x^k)\right\|^2 + 2(1 - \nu_k\beta_k)^2\left\|\xi^k\right\|^2 + 2\alpha_k^2 L^2\left\|b^k\right\|^2 \\
&\quad + 2\nu_k\beta_k\left\langle d^{k-1} - \nabla F(x^k),\, (1 - \nu_k\beta_k)\xi^k\right\rangle + \frac{\alpha_k^2}{1 - \nu_k\beta_k}L^2\left\|b^k\right\|^2. \quad (42)
\end{aligned}
$$

Taking expectation conditioned on $\{x^0, g^0, \ldots, x^{k-1}, d^{k-1}, x^k\}$, we have

$$
\begin{aligned}
\mathbf{E}_k\left[\left\|d^k - \nabla F(x^{k+1})\right\|^2\right] &\leq \nu_k\beta_k\left\|d^{k-1} - \nabla F(x^k)\right\|^2 + 2(1 - \nu_k\beta_k)^2\mathbf{E}_k\left[\left\|\xi^k\right\|^2\right] + 2\alpha_k^2 L^2\mathbf{E}_k\left[\left\|b^k\right\|^2\right] \\
&\quad + \frac{\alpha_k^2}{1 - \nu_k\beta_k}L^2\mathbf{E}_k\left[\left\|b^k\right\|^2\right]. \quad (43)
\end{aligned}
$$

To show that $\|d^{k-1} - \nabla F(x^k)\|^2$ is a convergent martingale, we prove the following lemma, similar to Ermoliev [5].

**Lemma 4.** *Assume we are given a sequence such that* $\mathbf{E}_k[X_{k+1}] \leq X_k + Y_k$, *where* $0 \leq X_k \leq C$ *and* $0 \leq Y_k \leq C$ *almost surely for some constant* $C$, *the random variables* $Y_k$ *are* $\mathcal{F}_k$*-measurable, and they satisfy* $\Sigma_{k=0}^{\infty} Y_k < \infty$ *almost surely. Then the sequence* $X_k$ *converges almost surely.*

*Proof.* We show that $Z_k = X_k + \sum_{k=0}^{\infty} Y_k$ is a convergent supermartingale. By Doob decomposition, $Z_k$ is a supermartingale if and only if the sequence

$$
A_k = \sum_{i=1}^{k} \mathbf{E}_{i-1}[Z_i - Z_{i-1}]
$$

satisfies $\mathbf{P}(A_{k+1} \leq A_k) = 1$ for all $k$ [38, section 12.11]. Here

$$
\mathbf{E}_{i-1}[Z_i - Z_{i-1}] = \mathbf{E}_{i-1}[X_k] - X_{k-1} - Y_{k-1},
$$

and we assumed this is non-positive. So $A_{k+1} \leq A_k$ almost surely. The upper bound on $X_k$ and the convergence of $\Sigma_{k=0}^{\infty} Y_k$ implies that the supermartingale $Z$ is in $\mathcal{L}^1$, so the sequence $\{Z_k\}$ converges almost surely by Doob's forward convergence theorem [38, chapter 11]. By the convergence of $\Sigma Y_k$, this in turn implies that the sequence $X_k$ converges almost surely. $\square$

We can apply the above lemma to show that $\left\|d^{k-1} - \nabla F(x^k)\right\|^2$ is a convergent semimartingale. Because the noise $\|\xi^k\|^2$ is uniformly bounded almost surely and $\|\nabla F(x^k)\| \le G$, $\|d^{k-1} - \nabla F(x^k)\|^2$ is uniformly bounded in $k$. The uniform bound on the noise $\|\xi^k\|^2$ also implies that $\|b^k\|^2$ is uniformly bounded in $k$. In the notation of the lemma, we have

$$Y_k = 2(1 - \nu_k\beta_k)^2 \mathbf{E}_k\left[\left\|\xi^k\right\|^2\right] + 2\alpha_k^2 L^2 \mathbf{E}_k\left[\left\|b^k\right\|^2\right] + \frac{\alpha_k^2}{1 - \nu_k\beta_k} L^2 \mathbf{E}_k\left[\left\|b^k\right\|^2\right].$$

Note that $Y_k \ge 0$. To show convergence of $\sum Y_k$, note that the uniform bounds imply

$$Y_k \le C\left((1 - \nu_k\beta_k)^2 + \alpha_k^2 + \frac{\alpha_k^2}{1 - \nu_k\beta_k}\right)$$

for suitably large $C$. Then convergence follows from the conditions on the sequences $(1 - \nu_k\beta_k)^2$, $\alpha_k^2$, and $\alpha_k^2/(1 - \nu_k\beta_k)$. This proves that $\|d^{k-1} - \nabla F(x^k)\|^2$ converges almost surely.

Summing up the two inequalities (41) and (43) gives

$$\mathbf{E}_k\left[F(x^{k+1}) + \left\|d^k - \nabla F(x^{k+1})\right\|^2\right]$$

$$\le \quad F(x^k) + (1 + \alpha_k)\nu_k\beta_k\left\|d^{k-1} - \nabla F(x^k)\right\|^2 - \frac{3\alpha_k}{4}\left\|\nabla F(x^k)\right\|^2$$

$$+2(1 - \nu_k\beta_k)^2 \mathbf{E}_k\left[\left\|\xi^k\right\|^2\right] + \frac{5}{2}\alpha_k^2 L^2 \mathbf{E}_k\left[\left\|b^k\right\|^2\right] + \frac{\alpha_k^2}{1 - \nu_k\beta_k} L^2 \mathbf{E}_k\left[\left\|b^k\right\|^2\right].$$

If $(1 + \alpha_k)\nu_k\beta_k \le 1$, then

$$\mathbf{E}_k\left[F(x^{k+1}) + \left\|d^k - \nabla F(x^{k+1})\right\|^2\right]$$

$$\le \quad F(x^k) + \left\|d^{k-1} - \nabla F(x^k)\right\|^2 - \frac{3\alpha_k}{4}\left\|\nabla F(x^k)\right\|^2$$

$$+2(1 - \nu_k\beta_k)^2 \mathbf{E}_k\left[\left\|\xi^k\right\|^2\right] + \frac{5}{2}\alpha_k^2 L^2 \mathbf{E}_k\left[\left\|b^k\right\|^2\right] + \frac{\alpha_k^2}{1 - \nu_k\beta_k} L^2 \mathbf{E}_k\left[\left\|b^k\right\|^2\right].$$

Rearranging terms, we get

$$\frac{3\alpha_k}{4}\left\|\nabla F(x^k)\right\|^2 \quad \le \quad F(x^k) + \left\|d^{k-1} - \nabla F(x^k)\right\|^2 - \mathbf{E}_k\left[F(x^{k+1}) + \left\|d^k - \nabla F(x^{k+1})\right\|^2\right]$$

$$+2(1 - \nu_k\beta_k)^2 \mathbf{E}_k\left[\left\|\xi^k\right\|^2\right] + \frac{5}{2}\alpha_k^2 L^2 \mathbf{E}_k\left[\left\|b^k\right\|^2\right] + \frac{\alpha_k^2}{1 - \nu_k\beta_k} L^2 \mathbf{E}_k\left[\left\|b^k\right\|^2\right].$$

Since $\alpha_k/(1 - \nu_k\beta_k) \to 0$, there exists $m$ such that $(1 + \alpha_k)\nu_k\beta_k \le 1$ for all $k \ge m$. Taking full expectation on both sides of the above inequality and summing up for all $k \ge m$, we obtain

$$\frac{3}{4}\sum_{k=m}^{\infty} \mathbf{E}\left[\alpha_k\left\|\nabla F(x^k)\right\|^2\right] \quad \le \quad \mathbf{E}\left[F(x^m) + \left\|d^{m-1} - \nabla F(x^m)\right\|^2\right] - F_*$$

$$+\sum_{k=m}^{\infty} 2C_\xi \mathbf{E}\left[\beta_k^2\right] + \sum_{k=m}^{\infty} \frac{5}{2}L^2 C_d \mathbf{E}\left[\alpha_k^2\right] + \sum_{k=m}^{\infty} L^2 C_d \mathbf{E}\left[\frac{\alpha_k^2}{\beta_k}\right]$$

$$\le \quad M + C\left(\sum_{k=m}^{\infty}(1 - \nu_k\beta_k)^2 + \sum_{k=m}^{\infty}\alpha_k^2 + \sum_{k=m}^{\infty}\frac{\alpha_k^2}{(1 - \nu_k\beta_k)}\right).$$

The right-hand side is bounded by assumption (the index $m$ is finite), so we have

$$\frac{3}{4}\sum_{k=m}^{\infty} \mathbf{E}\left[\alpha_k\left\|\nabla F(x^k)\right\|^2\right] < \infty.$$

This in turn implies that the series

$$\frac{3}{4}\sum_{k=m}^{\infty} \alpha_k\left\|\nabla F(x^k)\right\|^2 < \infty \qquad a.s.$$

So because $\sum_k \alpha_k = \infty$, there must be a subsequence $k_t$ with $\|\nabla F(x^{k_t})\|^2 \to 0$. This proves (7). $\quad\square$

# C   Local Convergence Rate Proofs

In this section we give a proof to Theorem 3 and it's generalized version, which we present below.

We will denote with $\lambda_i(A)$, $\rho(A)$ the $i$-th eigenvalue and spectral radius of the matrix $A$ respectively.

Let's recall the equations of deterministic QHM algorithm (6) with constant parameters $\alpha, \beta, \nu$:

$$
\begin{aligned}
d^k &= (1 - \beta)\nabla F(x^k) + \beta d^{k-1} \\
x^{k+1} &= x^k - \alpha \left[ (1 - \nu)\nabla F(x^k) + \nu d^k \right].
\end{aligned}
\tag{44}
$$

In this section we will assume that $d^0$ is initialized with zero vector.

Taking the gradient of the quadratic function $F(x) = x^T A x + b^T x + c$ and substituting it into (44) yields

$$
\begin{aligned}
d^k &= (1 - \beta)(Ax^k + b) + \beta d^{k-1} \\
x^{k+1} &= x^k - \alpha(1 - \nu\beta)(Ax^k + b) - \alpha\nu\beta d^{k-1}
\end{aligned}
$$

Plugging in $Ax_* = -b$ we get

$$
\begin{aligned}
d^k &= (1 - \beta)A(x^k - x_*) + \beta d^{k-1} \\
x^{k+1} - x_* &= x^k - x_* - \alpha(1 - \nu\beta)A(x^k - x_*) - \alpha\nu\beta d^{k-1}
\end{aligned}
$$

We can write the above two equations as

$$
\begin{bmatrix} d^k \\ x^{k+1} - x_* \end{bmatrix} = \begin{bmatrix} \beta I & (1-\beta)A \\ -\alpha\nu\beta I & I - \alpha(1 - \nu\beta)A \end{bmatrix} \begin{bmatrix} d^{k-1} \\ x^k - x_* \end{bmatrix} \triangleq T(\theta) \begin{bmatrix} d^{k-1} \\ x^k - x_* \end{bmatrix} = T^k(\theta, A) \begin{bmatrix} d^0 \\ x^0 - x_* \end{bmatrix},
$$

where $I$ denotes the $n \times n$ identity matrix and $\theta = \{\alpha, \beta, \nu\}$. It is known that the sequence of $T^k(\theta, A)$ converges to zero if and only if the spectral radius $\rho(T) < 1$. Moreover, Gelfand's Formula states that $\rho(T) = \lim_{k \to \infty} \left\| T^k \right\|^{\frac{1}{k}}$, which means that $\exists \{\epsilon_k\}_0^\infty$, $\lim_{k \to \infty} \epsilon_k = 0$ such that

$$
\left\| x^{k+1} - x_* \right\| \leq \left\| \begin{bmatrix} d^k \\ x^{k+1} - x_* \end{bmatrix} \right\| \leq \left\| T^k(\theta, A) \right\| \left\| \begin{bmatrix} d^0 \\ x^0 - x_* \end{bmatrix} \right\| \leq (\rho(T) + \epsilon_k)^k \left\| x^0 - x_* \right\|,
$$

Thus, the behavior of the algorithm is determined by the eigenvalues of $T(\theta)$. To find them, we will use a standard technique of changing basis. Let $A = Q\Lambda Q^T$ be an eigendecomposition of the matrix $A$. Then, multiplying $A$ with $Q$ and appropriate permutation matrix $P$[6] we get

$$
P \begin{bmatrix} Q & 0 \\ 0 & Q \end{bmatrix} T(\theta) \begin{bmatrix} Q & 0 \\ 0 & Q \end{bmatrix}^T P^T = \begin{bmatrix} T_1 & 0 & \cdots & 0 \\ 0 & T_2 & \cdots & 0 \\ \vdots & \vdots & \ddots & \vdots \\ 0 & 0 & \cdots & T_n \end{bmatrix}
$$

where $T_i \in \mathbb{R}^{2 \times 2}$ is defined as

$$
T_i = T_i(\theta, \lambda_i(A)) = \begin{bmatrix} \beta & (1-\beta)\lambda_i(A) \\ -\alpha\nu\beta & 1 - \alpha(1 - \nu\beta)\lambda_i(A) \end{bmatrix}
$$

Thus, to compute eigenvalues of $T$, it is enough to compute the eigenvalues of all matrices $T_i$.

We use the following Lemma to establish the region when $\rho(T_i) < 1$:

**Lemma 5.** *Let $\alpha > 0, \beta \in [0, 1), \nu \in [0, 1], \lambda_i(A) > 0$. Then*

$$
\rho(T_i(\theta)) < 1 \text{ if } \alpha < \frac{2(1 + \beta)}{\lambda_i(A)(1 + \beta(1 - 2\nu))}
$$

*Proof.* Let's denote with $\lambda$ eigenvalues of $T_i$. Let's also define $l \triangleq \lambda_i(A)$. Then, $\lambda$ satisfies the following equation:

$$(\beta - \lambda)(1 - \alpha(1 - \nu\beta)l - \lambda) + \alpha\nu\beta(1 - \beta)l = 0 \Leftrightarrow$$
$$\beta - \lambda - \beta\alpha(1 - \nu\beta)l + \lambda\alpha(1 - \nu\beta)l - \lambda\beta + \lambda^2 + \alpha\nu\beta l - \alpha\nu\beta^2 l = 0 \Leftrightarrow$$
$$\beta - \lambda - \beta\alpha l + \alpha\beta^2\nu l + \lambda\alpha l - \lambda\alpha\nu\beta l - \lambda\beta + \lambda^2 + \alpha\beta\nu l - \alpha\beta^2\nu l = 0 \Leftrightarrow$$
$$\lambda^2 - (1 - \alpha l + \alpha\nu\beta l + \beta)\lambda + \beta(1 - \alpha l + \alpha\nu l) = 0$$
$$D = (1 - \alpha l + \alpha\nu\beta l + \beta)^2 - 4\beta(1 - \alpha l + \alpha\nu l)$$

Let's denote by $S(A) = \{\alpha, \beta, \nu : A \text{ is true}\}$. The final convergence set
$$S(|\lambda| < 1) = S(|\lambda| < 1 \cap D \geq 0) \cup S(|\lambda| < 1 \cap D < 0)$$

Let's look at the case when $D \geq 0$. Then $S(|\lambda| < 1 \cap D \geq 0) = S(D \geq 0) \cap S(|\lambda_1| < 1) \cap S(|\lambda_2| < 1)$
$$\lambda_{1,2} = \frac{1 - \alpha l + \alpha\nu\beta l + \beta \pm \sqrt{D}}{2}$$

Let's look at $S(|\lambda_1| < 1) = S(\lambda_1 < 1) \cap S(\lambda_1 > -1)$
$$|\lambda_1| = \frac{\left|1 - \alpha l + \alpha\nu\beta l + \beta + \sqrt{D}\right|}{2} < 1 \Leftrightarrow \left|1 - \alpha l + \alpha\nu\beta l + \beta + \sqrt{D}\right| < 2 \Leftrightarrow$$
$$-2 < 1 - \alpha l + \alpha\nu\beta l + \beta + \sqrt{D} < 2 \Leftrightarrow$$
$$-3 + \alpha l - \alpha\nu\beta l - \beta < \sqrt{D} < 1 + \alpha l - \alpha\nu\beta l - \beta$$

Let's solve the second inequality: $S(\lambda_1 < 1)$. Since we are only interested in the case when $D \geq 0$ we get

$$\sqrt{D} < 1 + \alpha l - \alpha\nu\beta l - \beta \Leftrightarrow (1 - \alpha l + \alpha\nu\beta l + \beta)^2 - 4\beta(1 - \alpha l + \alpha\nu l) < (1 + \alpha l - \alpha\nu\beta l - \beta)^2 \Leftrightarrow$$
$$0 < 4\beta(1 - \alpha l + \alpha\nu l) + 4(\alpha l - \alpha\nu\beta l - \beta) \Leftrightarrow 0 < 4\alpha l(1 - \beta)$$

which is always satisfied.

Let's solve the first inequality:
$$S(\lambda_1 > -1) = S(-3 + \alpha l - \alpha\nu\beta l - \beta \leq 0) \cup S(-3 + \alpha l - \alpha\nu\beta l - \beta > 0 \cap -3 + \alpha l - \alpha\nu\beta l - \beta < \sqrt{D})$$

We can rewrite the first term as
$$S(-3 + \alpha l - \alpha\nu\beta l - \beta \leq 0) = S\left(\alpha \leq \frac{3 + \beta}{l(1 - \nu\beta)}\right)$$

Let's compute the second term

$$(\alpha l - \alpha\nu\beta l - \beta - 3)^2 < (\alpha l - \alpha\nu\beta l - \beta - 1)^2 - 4\beta(1 - \alpha l + \alpha\nu l) \Leftrightarrow$$
$$0 < 4(\alpha l - \alpha\nu\beta l - \beta) - 8 - 4\beta(1 - \alpha l + \alpha\nu l) \Leftrightarrow$$
$$0 < -8\beta - 8\alpha\nu\beta l + 4\alpha l(1 + \beta) - 8 \Leftrightarrow$$
$$2(1 + \beta) < \alpha l(1 + \beta - 2\beta\nu) \Leftrightarrow \alpha > \frac{2(1 + \beta)}{l(1 + \beta(1 - 2\nu))}$$

The last inequality is true since $1 + \beta(1 - 2\nu) > 0$. Thus
$$S(-3 + \alpha l - \alpha\nu\beta l - \beta > 0 \cap -3 + \alpha l - \alpha\nu\beta l - \beta < \sqrt{D}) =$$
$$S\left(\alpha > \frac{3 + \beta}{l(1 - \nu\beta)}\right) \cap S\left(\alpha > \frac{2(1 + \beta)}{l(1 + \beta(1 - 2\nu))}\right) =$$
$$S\left(\alpha > \frac{3 + \beta}{l(1 - \nu\beta)}\right)$$

Since $3 + \beta > 2(1 + \beta)$ and $l(1 - \nu\beta) \le l(1 + \beta(1 - 2\nu))$.

Therefore we have that $\lambda_1 > -1$ always holds and thus $|\lambda_1| < 1$ always holds.

Now we compute $S(|\lambda_2| < 1) = S(\lambda_2 < 1) \cap S(\lambda_2 > -1)$. The first term is

$$\lambda_2 = \frac{1 - \alpha l + \alpha\nu\beta l + \beta - \sqrt{D}}{2} < 1 \Leftrightarrow 1 - \alpha l + \alpha\nu\beta l + \beta - \sqrt{D} < 2 \Leftrightarrow -1 - \alpha l + \alpha\nu\beta l + \beta < \sqrt{D}$$

Which is always satisfied since

$$-1 - \alpha l + \alpha\nu\beta l + \beta = \beta - 1 + l\alpha(\nu\beta - 1) < 0$$

Let's compute the second term:

$$\lambda_2 = \frac{1 - \alpha l + \alpha\nu\beta l + \beta - \sqrt{D}}{2} > -1 \Leftrightarrow$$

$$1 - \alpha l + \alpha\nu\beta l + \beta - \sqrt{D} > -2 \Leftrightarrow \sqrt{D} < 3 - \alpha l + \alpha\nu\beta l + \beta \Leftrightarrow$$

$$(1 - \alpha l + \alpha\nu\beta l + \beta)^2 - 4\beta(1 - \alpha l + \alpha\nu l) < (3 - \alpha l + \alpha\nu\beta l + \beta)^2 \Leftrightarrow$$

$$-8 + 4(\alpha l - \alpha\nu\beta l - \beta) - 4\beta(1 - \alpha l + \alpha\nu l) < 0 \Leftrightarrow \alpha < \frac{2(1 + \beta)}{l(1 + \beta(1 - 2\nu))}$$

Thus we get

$$S(|\lambda_2| < 1) = S\left(\alpha < \frac{2(1 + \beta)}{l(1 + \beta(1 - 2\nu))}\right)$$

and therefore

$$S(|\lambda| < 1 \cap D > 0) = S(D > 0) \cap S\left(\alpha < \frac{2(1 + \beta)}{l(1 + \beta(1 - 2\nu))}\right)$$

Now let's move to the second case and compute $S(|\lambda| < 1 \cap D < 0)$. If $D < 0$ we have that $1 - \alpha l + \alpha\nu l > 0$ and then

$$|\lambda_{1,2}| = \frac{\left|1 - \alpha l + \alpha\nu\beta l + \beta \pm i\sqrt{-D}\right|}{2} = 0.5\sqrt{(1 - \alpha l + \alpha\nu\beta l + \beta)^2 - D} =$$

$$\sqrt{\beta(1 - \alpha l + \alpha\nu l)} < 1 \Leftrightarrow \alpha > \frac{\beta - 1}{l\beta(1 - \nu)}$$

which is always true, so

$$S(|\lambda| < 1 \cap D < 0) = S(D < 0)$$

Finally, let's find a simplified form of $S(D \ge 0)$ and $S(D < 0)$.

$$D = (1 - \alpha l + \alpha\nu\beta l + \beta)^2 - 4\beta(1 - \alpha l + \alpha\nu l) = (1 + \beta - \alpha l(1 - \nu\beta))^2 - 4\beta + 4\alpha l\beta - 4\alpha l\beta\nu$$

$$= 1 + 2\beta + \beta^2 - 2\alpha l(1 + \beta)(1 - \nu\beta) + \alpha^2 l^2 (1 - \nu\beta)^2 - 4\beta + 4\alpha l\beta - 4\alpha l\beta\nu$$

$$= \alpha^2 l^2 (1 - \nu\beta)^2 - 2\alpha l - 2\alpha l\beta + 2\alpha l\nu\beta + 2\alpha l\nu\beta^2 + 4\alpha l\beta - 4\alpha l\beta\nu + 1 - 2\beta + \beta^2$$

$$= \alpha^2 l^2 (1 - \nu\beta)^2 - 2\alpha l + 2\alpha l\beta - 2\alpha l\nu\beta + 2\alpha l\nu\beta^2 + (1 - \beta)^2$$

$$= \alpha^2 l^2 (1 - \nu\beta)^2 - 2\alpha l(1 - \beta + \nu\beta - \nu\beta^2) + (1 - \beta)^2$$

$$= \alpha^2 l^2 (1 - \nu\beta)^2 - 2\alpha l(1 - \beta)(1 + \nu\beta) + (1 - \beta)^2$$

Let's denote the discriminant of that equation (divided by 4) with respect to $\alpha l$ as $D_1$:

$$D_1 = (1-\beta)^2(1+\nu\beta)^2 - (1-\nu\beta)^2(1-\beta)^2 = 4\nu\beta(1-\beta)^2 \geq 0$$

$$\alpha l_{1,2} = \frac{(1-\beta)(1+\nu\beta) \pm 2(1-\beta)\sqrt{\nu\beta}}{(1-\nu\beta)^2} = \frac{(1-\beta)(1+\nu\beta \pm 2\sqrt{\nu\beta})}{(1-\nu\beta)^2} = \frac{(1-\beta)(1 \pm 2\sqrt{\nu\beta})^2}{(1-\nu\beta)^2}$$

Therefore

$$S(D \geq 0) = S\left(\alpha \geq \frac{(1-\beta)(1+\sqrt{\nu\beta})^2}{l(1-\nu\beta)^2} \cup \alpha \leq \frac{(1-\beta)(1-\sqrt{\nu\beta})^2}{l(1-\nu\beta)^2}\right)$$

$$S(D < 0) = S\left(\alpha \in \left[\frac{(1-\beta)(1-\sqrt{\nu\beta})^2}{l(1-\nu\beta)^2}, \frac{(1-\beta)(1+\sqrt{\nu\beta})^2}{l(1-\nu\beta)^2}\right]\right)$$

(45)

Now, notice that
$$\frac{(1-\beta)(1+\sqrt{\nu\beta})^2}{l(1-\nu\beta)^2} = \frac{(1+\sqrt{\nu\beta})^2}{\frac{l(1-\nu\beta)^2}{1-\beta}} < \frac{2(1+\beta)}{l(1+\beta(1-2\nu))}$$

since $(1+\sqrt{\nu\beta})^2 < 2(1+\beta)$ (left side is less than 2 and right side is greater than 2) and also

$$\frac{l(1-\nu\beta)^2}{1-\beta} \geq l(1+\beta(1-2\nu)) \Leftrightarrow 1 - 2\nu\beta + \nu^2\beta^2 \geq 1 - \beta + \beta - \beta^2 - 2\nu\beta + 2\nu\beta^2 \Leftrightarrow \beta^2(1-\nu)^2 \geq 0$$

Thus overall we get that

$$S(|\lambda| < 1 \cap D \geq 0) = S(D \geq 0) \cap S\left(\alpha < \frac{2(1+\beta)}{l(1+\beta(1-2\nu))}\right) =$$

$$= S\left(\alpha < \frac{2(1+\beta)}{l(1+\beta(1-2\nu))}\right) \setminus S(D < 0) \Rightarrow$$

$$S(|\lambda| < 1) = S(|\lambda| < 1 \cap D \geq 0) \cup S(|\lambda| < 1 \cap D < 0)$$

$$= S\left(\alpha < \frac{2(1+\beta)}{l(1+\beta(1-2\nu))}\right) \setminus S(D < 0) \cup S(D < 0) =$$

$$= S\left(\alpha < \frac{2(1+\beta)}{l(1+\beta(1-2\nu))}\right)$$

$\square$

Now, let's establish a precise equation for the spectral radius $\rho(T_i)$.

**Lemma 6.** *Let $\alpha > 0, \beta \in [0, 1), \nu \in [0, 1], \lambda_i(A) > 0$. Let's define $l \triangleq \lambda_i(A)$ and*
$$C_1 = 1 - \alpha l + \alpha l\nu\beta + \beta$$
$$C_2 = \beta(1 - \alpha l + \alpha l\nu)$$

*Then*

$$r(\theta, l) = \rho(T_i(\theta)) = \begin{cases} 0.5\left(\sqrt{C_1^2 - 4C_2} + C_1\right) & \text{if } C_1 \geq 0, C_1^2 - 4C_2 \geq 0 \\ 0.5\left(\sqrt{C_1^2 - 4C_2} - C_1\right) & \text{if } C_1 < 0, C_1^2 - 4C_2 \geq 0 \\ \sqrt{C_2} & \text{if } C_1^2 - 4C_2 < 0 \end{cases}$$

*In addition, $r(\theta, l)$ is non-increasing as a function of $l$ for $0 < l < \frac{1-\beta}{\alpha(1-\sqrt{\nu\beta})^2}$ and is non-decreasing for $l > \frac{1-\beta}{\alpha(1-\sqrt{\nu\beta})^2}$.*

*Proof.* Following derivations from the proof of Lemmas 5 we get

$$r(\theta, l) = \begin{cases} \max\left\{0.5\left|C_1 + \sqrt{C_1^2 - 4C_2}\right|, 0.5\left|C_1 - \sqrt{C_1^2 - 4C_2}\right|\right\} & \text{if } C_1^2 - 4C_2 \geq 0 \\ \sqrt{C_2} & \text{if } C_1^2 - 4C_2 < 0 \end{cases}$$

Considering 4 cases for different signs of $C_1$ and $C_2$, the first statement of the Lemma immediately follows. To prove the second statement, let's define the following 3 points:

$$p_1 = \frac{1 - \beta}{\alpha(1 + \sqrt{\nu\beta})^2}, \quad p_2 = \frac{1 - \beta}{\alpha(1 - \sqrt{\nu\beta})^2}, \quad p_3 = \frac{1 + \beta}{\alpha(1 - \nu\beta)}$$

From equation (45) and definition of $C_1$ we get that

$$C_1 \geq 0 \Leftrightarrow l \leq p_3$$

$$C_1^2 - 4C_2 \geq 0 \Leftrightarrow l \leq p_1 \text{ or } l \geq p_2$$

and it is easy to check that if $\beta \geq \nu \Rightarrow p_1 \leq p_2 \leq p_3$ and if $\beta < \nu \Rightarrow p_1 \leq p_3 \leq p_2$. Moreover, both $C_1(l)$ and $C_2(l)$ are non-increasing function of $l$, and $C_1^2(l) - 4C_2(l)$ is non-increasing when $l \leq p_2$ and non-decreasing when $l \geq p_1$.

Let's first prove the second statement of the Lemma for the case when $\beta < \nu$. In that case, when $l < p_1$ the function is non-increasing, since both $C_1(l)$ and $C_1^2(l) - 4C_2(l)$ are non-increasing. When $p_1 \leq l \leq p_2$, the function is non-increasing, because $C_2(l)$ is non-increasing. Finally, when $l > p_2$, the function is non-decreasing, because both $C_1^2(l) - 4C_2(l)$ and $-C_1(l)$ are non-decreasing.

When $\beta \geq \nu$, the same reasoning applies, but we additionally need to prove that the function is non-decreasing when $p_2 \leq l \leq p_3$. In that case $r(\theta, l) = 0.5(\sqrt{C_1^2(l) - 4C_2(l)} + C_1(l))$. Taking the derivative of $r$ with respect to $l$ we get

$$\frac{\partial r}{\partial l} = \frac{2\alpha\beta(1 - \nu) - \alpha(1 - \nu\beta)\left(\sqrt{C_1^2(l) - 4C_2(l)} + C_1(l)\right)}{2\sqrt{C_1^2(l) - 4C_2(l)}}$$

Let's show that this derivative is always non-negative when $l \geq p_2$

$$\frac{2\alpha\beta(1 - \nu) - \alpha(1 - \nu\beta)\left(\sqrt{C_1^2(l) - 4C_2(l)} + C_1(l)\right)}{2\sqrt{C_1^2(l) - 4C_2(l)}} \geq 0 \Leftrightarrow$$

$$2\beta(1 - \nu) - (1 - \nu\beta)\left(\sqrt{C_1^2(l) - 4C_2(l)} + C_1(l)\right) \geq 0 \Leftrightarrow$$

$$4\beta(1 - \nu)^2 - 4\beta(1 - \nu)(1 - \nu\beta)C_1(l) + (1 - \nu\beta)C_1^2(l) \geq (C_1^2(l) - 4C_2(l))(1 - \nu\beta)^2 \Leftrightarrow$$

$$\beta(1 - \nu)^2 - \beta(1 - \nu)(1 - \nu\beta)C_1(l) + C_2(l)(1 - \nu\beta)^2 \geq 0 \Leftrightarrow$$

$$\frac{-\beta^2(1 - \nu^2) + (1 - \nu)(1 - \nu\beta)(1 + \beta) - \beta(1 - \nu\beta)^2}{\alpha(1 - \nu)(1 - \nu\beta)^2(1 - \beta)} \leq l \Leftrightarrow$$

$$-\frac{(1 - \beta)^2\nu\beta}{\alpha(1 - \nu)(1 - \nu\beta)^2(1 - \beta)} \leq l$$

which is always true since left side is less than zero.

$\square$

The last thing that we need in order to prove Theorem 3 is given by the following Lemma:

**Lemma 7.** *Let $\mu \leq \min_i \lambda_i(A)$ and $L \geq \max_i \lambda_i(A)$. Then*

$$\rho(T(\theta)) \leq R(\theta, \mu, L) = \max(r(\theta, \mu), r(\theta, L))$$

*In addition, the minimal spectral radius with respect to $\theta$ depends on $\mu$ and $L$ only through $\kappa$, i.e. $\min_\theta R(\theta, \mu, L) \triangleq R^*(\kappa)$*

*Proof.* To prove this first statement of the Lemma, let's notice that by definition

$$\rho(T(\theta)) = \max_i \rho(T_i(\theta))$$

But Lemma 6 states that $\rho(T_i(\theta))$ is first non-increasing and then non-decreasing with respect to the eigenvalues of $A$. Thus, the maximum can only be achieved on the boundaries, which are precisely equal to or smaller than $r(\theta, \mu)$ and $r(\theta, L)$.

Let's prove the second statement of the Lemma by contradiction. Let's assume that the optimal rate does in fact depend on $\mu$ and $L$ not only through $\kappa$. That means that $\exists \mu_1, L_1, \mu_2, L_2$, such that $L_1/\mu_1 = L_2/\mu_2$, but $\min_\theta R(\theta, \mu_1, L_1) \neq \min_\theta R(\theta, \mu_2, L_2)$. Let's consider the optimal rates if the function $f$ is divided by $\mu_1$ for the first case and by $\mu_2$ for the second. In that case, $\min_\theta R(\theta, 1, L_1/\mu_1) = \min_\theta R(\theta, 1, L_2/\mu_2)$. But on the other hand, they can't be equal, since we have that $\min_\theta R(\theta, 1, L_1/\mu_1) = \min_\theta R(\theta, \mu_1, L_1)$ and $\min_\theta R(\theta, 1, L_2/\mu_2) = \min_\theta R(\theta, \mu_2, L_2)$, because multiplying learning rate by $\mu_1$ for the first case and by $\mu_2$ for the second yields exactly the same sequence of iterates and thus the optimal rate can't change. $\square$

Now we are ready to prove Theorem 3. We restate it below for convenience

**Theorem 3.** *Let's denote $\theta = \{\alpha, \beta, \nu\}$. For any function $F(x) = x^T A x + b^T x + c$ that satisfies $\mu \leq \lambda_i(A) \leq L$ for all $i = 1, \ldots, n$ and any $x^0$, the deterministic QHM algorithm $z^{k+1} = Tz^k$ satisfies*

$$\left\| x^k - x_* \right\| \leq (R(\theta, \mu, L) + \epsilon_k)^k \left\| x^0 - x_* \right\|,$$

*where $x_* = \arg\min_x F(x)$, $\lim_{k \to \infty} \epsilon_k = 0$ and $R(\theta, \mu, L) = \rho(T)$, which can be characterized as*

$$R(\theta, \mu, L) = \max\{r(\theta, \mu), r(\theta, L)\}, \quad \text{where}$$

$$r(\theta, \lambda) = \begin{cases} 0.5 \left( \sqrt{C_1(\lambda)^2 - 4C_2(\lambda)} + C_1(\lambda) \right) & \text{if } C_1(\lambda) \geq 0, C_1(\lambda)^2 - 4C_2(\lambda) \geq 0, \\ 0.5 \left( \sqrt{C_1(\lambda)^2 - 4C_2(\lambda)} - C_1(\lambda) \right) & \text{if } C_1(\lambda) < 0, C_1(\lambda)^2 - 4C_2(\lambda) \geq 0, \\ \sqrt{C_2(\lambda)} & \text{if } C_1(\lambda)^2 - 4C_2(\lambda) < 0, \end{cases}$$

$$C_1(\lambda, \theta) = 1 - \alpha\lambda + \alpha\lambda\nu\beta + \beta,$$
$$C_2(\lambda, \theta) = \beta(1 - \alpha\lambda + \alpha\lambda\nu).$$

*To ensure $R(\theta, \mu, L) < 1$, the parameters $\alpha, \beta, \nu$ must satisfy the following constraints:*

$$0 < \alpha < \frac{2(1 + \beta)}{L(1 + \beta(1 - 2\nu))}, \qquad 0 \leq \beta < 1, \qquad 0 \leq \nu \leq 1.$$

*In addition, the optimal rate depends only on $\kappa$: $\min_\theta R(\theta, \mu, L)$ is a function of only $\kappa$.*

*Proof.* Lemma 6 and Lemma 7 immediately give the first statement of the Theorem. One can also get the bound on the function values by using definition of the Lipschitz continuous gradient:

$$F(x^k) - F(x_*) \leq \nabla F(x_*)^T (x^k - x_*) + \frac{L}{2} \left\| x^k - x_* \right\|^2 = \frac{L}{2} \left\| x^k - x_* \right\|^2$$

Finally, to get the stability region, we apply Lemma 5 and notice that $\lambda_i(A) \leq L \ \forall i$. $\square$

To generalize this result, let's define the following class of functions

**Definition 1.** $\mathcal{F}^1_{\mu,L}$ *is the class of all functions $F : \mathbb{R}^n \to \mathbb{R}$ that are continuously differentiable, strongly convex with parameter $\mu$ and have Lipschitz continuous gradient with parameter L. We will denote the condition number of F as $\kappa = L/\mu$.*

Then, Theorem 3 can be generalized to any function $F \in \mathcal{F}^1_{\mu,L}$ in the following way:

**Theorem 6.** *Let's denote $\theta = \{\alpha, \beta, \nu\}$. For any function $F \in \mathcal{F}^1_{\mu,L}$ that is additionally twice differentiable at the point $x_* = \arg\min_x F(x)$, deterministic QHM algorithm locally converges to $x_*$ with linear rate, from any initialization $x^0$ sufficiently close to $x_*$.*

*Precisely, for any $\epsilon \in [0, 1 - R(\theta, \mu, L)) \; \exists \, \delta > 0$ and $c \geq 0$, such that $\forall k \geq 0$ the following holds*

$$\left\| x^k - x_* \right\| \leq c \left( R(\theta, \mu, L) + \epsilon \right)^k$$

$$F(x^k) - F(x_*) \leq \frac{c^2 L}{2} \left( R(\theta, \mu, L) + \epsilon \right)^{2k}$$

$$R(\theta, \mu, L) = \max \left\{ r(\theta, \mu), r(\theta, L) \right\}$$

$$r(\theta, \lambda) = \begin{cases} 0.5 \left( \sqrt{C_1(\lambda)^2 - 4C_2(\lambda)} + C_1(\lambda) \right) & \text{if } C_1(\lambda) \geq 0, C_1(\lambda)^2 - 4C_2(\lambda) \geq 0 \\ 0.5 \left( \sqrt{C_1(\lambda)^2 - 4C_2(\lambda)} - C_1(\lambda) \right) & \text{if } C_1(\lambda) < 0, C_1(\lambda)^2 - 4C_2(\lambda) \geq 0 \\ \sqrt{C_2(\lambda)} & \text{if } C_1(\lambda)^2 - 4C_2(\lambda) < 0 \end{cases}$$

$$C_1(\lambda, \theta) = 1 - \alpha\lambda + \alpha\lambda\nu\beta + \beta$$

$$C_2(\lambda, \theta) = \beta(1 - \alpha\lambda + \alpha\lambda\nu)$$

*if $\left\| x^0 - x_* \right\| \leq \delta$ and $\alpha, \beta, \nu$ satisfy the following constraints:*

$$0 < \alpha < \frac{2(1 + \beta)}{L(1 + \beta(1 - 2\nu))}, 0 \leq \beta < 1, 0 \leq \nu \leq 1$$

*In addition, the optimal rate depends on $\mu$ and $L$ only through $\kappa$, i.e. $\min_\theta R(\theta, \mu, L) \triangleq R^*(\kappa)$.*

*Proof.* To prove this result we apply Lyapunov's method (see e.g. Chapter 2, Theorem 1 of [28]) to the QHM equations. The proof is then identical to the proof of Theorem 3, with matrix $A$ replaced by $\nabla^2 F(x_*)$. $\qquad\square$

## D   Numerical Evaluation of the Convergence Rate

In this section we provide details on the numerical evaluation of the local convergence rate of QHM. We need to numerically estimate the following function

$$R^*(\nu, \kappa) = \min_{\alpha, \beta} \max \{ r(\alpha, \beta, \nu, \mu), r(\alpha, \beta, \nu, L) \}$$

From Lemma 6 (Appendix C) we know that $r(\alpha, \beta, \nu, l)$ is a non-increasing function of $l$ until some point and non-decreasing after. Also note that in fact dependence of $r$ on $\alpha$ is the same as on $l$, since they only appear in formulas as a product $\alpha l$. Thus, it is easy to see that for optimal $\alpha$ we will have

$$r(\alpha, \beta, \nu, \mu) = r(\alpha, \beta, \nu, L), \tag{46}$$

because otherwise $\alpha$ could be changed to decrease the value of the maximum.

Thus, to find optimal $\alpha$ for fixed $\beta, \nu$, we can solve equation (46) for $\alpha$ using binary search (with precision set to $10^{-8}$). To find optimal $\beta$ or $\nu$ we just use grid search (with grid size equal to $10^3$) on $[0, 1 - 10^{-5}]$ for $\beta$ and $[0, 1]$ for $\nu$.

To numerically verify that the dependence of the optimal rate on $\nu$ is monotonic, we run this procedure for $10^3$ values of $\kappa$ which are sampled (on a uniform grids) in the following way: 100 values on $[1, 10]$, 100 values on $[10, 100]$, 100 values on $[100, 1000]$, 150 values on $[10^3, 10^4]$, 150 values on $[10^4, 10^5]$, 200 values on $[10^5, 10^6]$, 200 values on $[10^6, 10^7]$. All experiments were run in parallel using GNU Parallel Command-Line Tool [35].

Since rate estimation is non-exact, it happens sometimes that very close points $\nu$ show non-monotonic rate dependence, but it is always the case that the rate is approximately non-increasing in $\nu$. Precisely, we verify that the following condition holds for all estimated values of $\kappa$:

$$\bar{R}^*(\nu_{i+10}, \kappa) - \bar{R}^*(\nu_i, \kappa) < 10^{-3} \; \forall i = 1, \ldots, 990$$

where $\bar{R}^*$ is estimated rate and $\nu_i$ is i-th sample of $\nu$. Figure 5 shows the dependence of $R^*(\nu, \kappa)$ on $\nu$ for different values of $\kappa$.

Figure 5: This Figure shows the dependence of optimal (across $\alpha, \beta$) local convergence rate on $\nu$ for QHM algorithm across different values of condition number $\kappa$. Note that $R^*(\nu, \kappa)$ always shows monotonic non-decreasing dependence on $\nu$.

## E    Stationary Distribution Proofs

In this section we present proofs of Theorems 4, 5. We will restate the combined statement of both theorems below for convenience.

**Theorems 4, 5.** *Suppose $F(x) = \frac{1}{2} x^T A x$, where $A$ is symmetric positive definite matrix. The stochastic gradients satisfy $g^k = \nabla F(x^k) + \xi$, where $\xi$ is a random vector independent of $x^k$ with zero mean $\mathbf{E}[\xi] = 0$ and covariance matrix $\mathbf{E}[\xi \xi^T] = \Sigma_\xi$. Also suppose the parameteres $\alpha, \beta, \nu$ satisfy (13), then QHM algorithm (6), equivalently (10) in this case, converges to stationary distribution satisfying*

$$A\Sigma_x + \Sigma_x A = \alpha A \Sigma_\xi + O(\alpha^2)$$

$$\mathbf{tr}(A\Sigma_x) = \frac{\alpha}{2}\mathbf{tr}(\Sigma_\xi) + \frac{\alpha^2}{4}\left(1 + \frac{2\nu\beta}{1-\beta}\left[\frac{2\nu\beta}{1+\beta} - 1\right]\right)\mathbf{tr}(A\Sigma_\xi) + O(\alpha^3)$$

*Consequently, when $\nu = 0$ (SGD), $\Sigma_x$ satisfies*

$$\mathbf{tr}(A\Sigma_x) = \frac{\alpha}{2}\mathbf{tr}(\Sigma_\xi) + \frac{\alpha^2}{4}\mathbf{tr}(A\Sigma_\xi) + O(\alpha^3)$$

*When $\nu = 1$ (SHB), $\Sigma_x$ satisfies*

$$\mathbf{tr}(A\Sigma_x) = \frac{\alpha}{2}\mathbf{tr}(\Sigma_\xi) + \frac{\alpha^2}{4}\frac{1-\beta}{1+\beta}\mathbf{tr}(A\Sigma_\xi) + O(\alpha^3)$$

*When $\nu = \beta$ (NAG), $\Sigma_x$ satisfies*

$$\mathbf{tr}(A\Sigma_x) = \frac{\alpha}{2}\mathbf{tr}(\Sigma_\xi) + \frac{\alpha^2}{4}\left(1 - \frac{2\beta^2(1+2\beta)}{1+\beta}\right)\mathbf{tr}(A\Sigma_\xi) + O(\alpha^3)$$

*Proof.* We consider the behavior of QHM with constant $\alpha$, $\beta$, and $\nu$, described in (47).

$$\begin{aligned} d^k &= (1-\beta)g^k + \beta d^{k-1} \\ x^{k+1} &= x^k - \alpha\left[(1-\nu)g^k + \nu d^k\right]. \end{aligned} \tag{47}$$

Under assumptions of Theorems 4, 5 we have that stochastic gradient $g^k$ is generated as

$$g^k = \nabla F(x^k) + \xi^k = Ax^k + \xi^k, \tag{48}$$

where the noise $\xi^k$ is independent of $x^k$, has zero mean and constant covariance matrix. More explicitly, for all $k \geq 0$,

$$\mathbf{E}[\xi^k] = 0, \qquad \mathbf{E}[\xi^k(\xi^k)^T] = \Sigma_\xi,$$

where $\Sigma_\xi$ is a constant covariance matrix. Substituting the expression of $g^k$ in (48) into (47) yields

$$d^k = (1-\beta)g^k + \beta d^{k-1} = (1-\beta)Ax^k + (1-\beta)\xi^k + \beta d^{k-1},$$

$$x^{k+1} = x^k - \alpha\nu d^k - \alpha(1-\nu)g^k = x^k - \alpha(1-\nu\beta)Ax^k - \alpha\nu\beta d^{k-1} - \alpha(1-\nu\beta)\xi^k.$$

We can write the above two equations as

$$\begin{bmatrix} d^k \\ x^{k+1} \end{bmatrix} = \begin{bmatrix} \beta I & (1-\beta)A \\ -\alpha\nu\beta I & I - \alpha(1-\nu\beta)A \end{bmatrix} \begin{bmatrix} d^{k-1} \\ x^k \end{bmatrix} + \begin{bmatrix} (1-\beta)I \\ -\alpha(1-\nu\beta)I \end{bmatrix} \xi^k, \tag{49}$$

where $I$ denotes the $n \times n$ identity matrix.

Let $L > 0$ be the largest eigenvalue of $A$. From Theorem 3 we know that under the conditions (13) the dynamical system (49) is stable, i.e., the spectral radius of the matrix

$$\begin{bmatrix} \beta I & (1-\beta)A \\ -\alpha\nu\beta I & I - \alpha(1-\nu\beta)A \end{bmatrix}$$

is smaller than one.

To simplify notation, we rewrite Equation (49) as

$$z^{k+1} = z^k - Bz^k + C\xi^k, \tag{50}$$

where

$$z^k = \begin{bmatrix} d^{k-1} \\ x^k \end{bmatrix}, \qquad B = \begin{bmatrix} (1-\beta)I & -(1-\beta)A \\ \alpha\nu\beta I & \alpha(1-\nu\beta)A \end{bmatrix}, \qquad C = \begin{bmatrix} (1-\beta)I \\ -\alpha(1-\nu\beta)I \end{bmatrix}.$$

As $k \to \infty$, the effect of the initial point $z^0$ dies out and the covariance matrix of the state $z^k$ becomes constant. Let

$$\Sigma_z = \begin{bmatrix} \Sigma_d & \Sigma_{dx} \\ \Sigma_{xd} & \Sigma_x \end{bmatrix} \triangleq \lim_{k\to\infty} \mathbf{E}\left[z^k(z^k)^T\right] = \lim_{k\to\infty} \begin{bmatrix} \mathbf{E}\left[d^{k-1}(d^{k-1})^T\right] & \mathbf{E}\left[d^{k-1}(x^k)^T\right] \\ \mathbf{E}\left[x^k(d^{k-1})^T\right] & \mathbf{E}\left[x^k(x^k)^T\right] \end{bmatrix}.$$

Then using the linear dynamics (50) and the assumption that $\{\xi^k\}$ is i.i.d. and has zero mean, we obtain

$$B\Sigma_z + \Sigma_z B^T - B\Sigma_z B^T = C\Sigma_\xi C^T.$$

Following the partition of $\Sigma_z$, we partition the above matrix equation into 2 by 2 blocks and obtain

$(1,1):$ $\quad (1-\beta^2)\Sigma_d - \beta(1-\beta)(A\Sigma_{xd} + \Sigma_{dx}A) - (1-\beta)^2 A\Sigma_x A = (1-\beta)^2\Sigma_\xi,$ $\tag{51}$

$(1,2):$ $\quad \alpha\nu\beta^2\Sigma_d + (1-\beta)\Sigma_{dx} - (1-\beta)A\Sigma_x + \alpha\nu\beta(1-\beta)A\Sigma_{xd} + \alpha\beta(1-\nu\beta)\Sigma_{dx}A$
$\quad +\alpha(1-\nu\beta)(1-\beta)A\Sigma_x A = -\alpha(1-\nu\beta)(1-\beta)\Sigma_\xi,$ $\tag{52}$

$(2,1):$ $\quad \alpha\nu\beta^2\Sigma_d + (1-\beta)\Sigma_{xd} - (1-\beta)\Sigma_x A + \alpha\beta(1-\nu\beta)A\Sigma_{xd} + \alpha\nu\beta(1-\beta)\Sigma_{dx}A$
$\quad +\alpha(1-\nu\beta)(1-\beta)A\Sigma_x A = -\alpha(1-\nu\beta)(1-\beta)\Sigma_\xi,$ $\tag{53}$

$(2,2):$ $\quad -(\alpha\nu\beta)^2\Sigma_d + \alpha\nu\beta(\Sigma_{xd} + \Sigma_{dx}) + \alpha(1-\nu\beta)(A\Sigma_x + \Sigma_x A)$
$\quad -\alpha^2\nu\beta(1-\nu\beta)(A\Sigma_{xd} + \Sigma_{dx}A) - \alpha^2(1-\nu\beta)^2 A\Sigma_x A = \alpha^2(1-\nu\beta)^2\Sigma_\xi.$ $\tag{54}$

Or, letting $V$ be the column block matrix with entries $[\Sigma_d, \Sigma_{xd}, \Sigma_{dx}, A\Sigma_x, \Sigma_x A, A\Sigma_{xd}, \Sigma_{dx}A, A\Sigma_x A]$, and defining symbolically $U$ to be the block matrix with coefficients:

$$\begin{bmatrix} (1-\beta^2) & 0 & 0 & 0 & 0 & -\beta(1-\beta) & -\beta(1-\beta) & -(1-\beta)^2 \\ \alpha\nu\beta^2 & 0 & (1-\beta) & -(1-\beta) & 0 & \alpha\nu\beta(1-\beta) & \alpha\beta(1-\nu\beta) & \alpha(1-\nu\beta)(1-\beta) \\ \alpha\nu\beta^2 & (1-\beta) & 0 & 0 & -(1-\beta) & \alpha\beta(1-\nu\beta) & \alpha\nu\beta(1-\beta) & \alpha(1-\nu\beta)(1-\beta) \\ -(\alpha\nu\beta)^2 & \alpha\nu\beta & \alpha\nu\beta & \alpha(1-\nu\beta) & \alpha(1-\nu\beta) & -\alpha^2\nu\beta(1-\nu\beta) & -\alpha^2\nu\beta(1-\nu\beta) & -\alpha^2(1-\nu\beta)^2, \end{bmatrix} \tag{55}$$

(each block is an $n \times n$ identity matrix), we have

$$UV = \begin{bmatrix} (1-\beta)^2 \\ -\alpha(1-\nu\beta)(1-\beta) \\ -\alpha(1-\nu\beta)(1-\beta) \\ \alpha^2(1-\nu\beta)^2 \end{bmatrix} \Sigma_\xi. \tag{56}$$

Next we use combinations of the above equations to obtain simplified relations: First, we can do

$$\frac{\alpha^2(1 - \nu\beta)^2}{(1 - \beta)^2}(1, 1) + \frac{\alpha(1 - \nu\beta)}{1 - \beta}[(1, 2) + (2, 1)] + (2, 2)$$

to get

$$\frac{\alpha(1 + \beta - 2\nu\beta)}{1 - \beta}\Sigma_d + \Sigma_{xd} + \Sigma_{dx} = 0.$$

We take the following asymptotic expansion of $\Sigma_z$:

$$\begin{bmatrix} \Sigma_d & \Sigma_{dx} \\ \Sigma_{xd} & \Sigma_x \end{bmatrix} = \begin{bmatrix} \Sigma_d^{(0)} + \alpha\Sigma_d^{(1)} + \alpha^2\Sigma_d^{(2)}/2 & \alpha\Sigma_{dx}^{(1)} + \alpha^2\Sigma_{dx}^{(2)}/2 \\ \alpha\Sigma_{xd}^{(1)} + \alpha^2\Sigma_{xd}^{(2)}/2 & \alpha\Sigma_x^{(1)} + \alpha^2\Sigma_x^{(2)}/2 \end{bmatrix}. \tag{57}$$

Here, we explicitly write the zero'th order of $(\Sigma_{dx}, \Sigma_{xd}, \Sigma_x)$ to be zero. This can be easily proved from (51)-(54).

The zero'th order term of (51) gives

$$(1 - \beta^2)\Sigma_d^{(0)} = (1 - \beta)^2\Sigma_\xi. \tag{58}$$

The first order term of (52) (and (53)) gives

$$\nu\beta^2\Sigma_d^{(0)} + (1 - \beta)(\Sigma_{dx}^{(1)} - A\Sigma_x^{(1)}) = -(1 - \nu\beta)(1 - \beta)\Sigma_\xi, \tag{59}$$

$$\nu\beta^2\Sigma_d^{(0)} + (1 - \beta)(\Sigma_{xd}^{(1)} - \Sigma_x^{(1)}A) = -(1 - \nu\beta)(1 - \beta)\Sigma_\xi. \tag{60}$$

The second order term of (54) gives

$$\nu\beta(\Sigma_{xd}^{(1)} + \Sigma_{dx}^{(1)}) + (1 - \nu\beta)(A\Sigma_x^{(1)} + \Sigma_x^{(1)}A) = \nu^2\beta^2\Sigma_d^{(0)} + (1 - \nu\beta)^2\Sigma_\xi. \tag{61}$$

From (58) we solve

$$\Sigma_d^{(0)} = \frac{1 - \beta}{1 + \beta}\Sigma_\xi,$$

and from (58), (59) and (60), we solve

$$\Sigma_{xd}^{(1)} + \Sigma_{dx}^{(1)} = A\Sigma_x^{(1)} + \Sigma_x^{(1)}A - \frac{2(1 + \beta - \nu\beta)}{1 + \beta}\Sigma_\xi. \tag{62}$$

After plugging them into (61), we obtain

$$A\Sigma_x^{(1)} + \Sigma_x^{(1)}A = \Sigma_\xi, \tag{63}$$

thus

$$A\Sigma_x + \Sigma_x A = \alpha\Sigma_\xi + O(\alpha^2),$$

which concludes the proof of Theorem 4.

Let's now extend this result to the second-order in $\alpha$. The first order term of (51) gives

$$(1 + \beta)\Sigma_d^{(1)} = \beta(A\Sigma_{xd}^{(1)} + \Sigma_{dx}^{(1)}A) + (1 - \beta)A\Sigma_x^{(1)}A. \tag{64}$$

The second order term of (52) (and (53)) gives

$$\nu\beta^2\Sigma_d^{(1)} + \frac{1 - \beta}{2}(\Sigma_{dx}^{(2)} - A\Sigma_x^{(2)}) = -\beta(1 - \nu\beta)\Sigma_{dx}^{(1)}A - \nu\beta(1 - \beta)A\Sigma_{xd}^{(1)} - (1 - \nu\beta)(1 - \beta)A\Sigma_x^{(1)}A, \tag{65}$$

$$\nu\beta^2\Sigma_d^{(1)} + \frac{1 - \beta}{2}(\Sigma_{xd}^{(2)} - \Sigma_x^{(2)}A) = -\beta(1 - \nu\beta)A\Sigma_{xd}^{(1)} - \nu\beta(1 - \beta)\Sigma_{dx}^{(1)}A - (1 - \nu\beta)(1 - \beta)A\Sigma_x^{(1)}A. \tag{66}$$

The third order term of (54) gives

$$-\nu^2\beta^2\Sigma_d^{(1)} + \frac{\nu\beta}{2}(\Sigma_{xd}^{(2)} + \Sigma_{dx}^{(2)}) + \frac{1 - \nu\beta}{2}(A\Sigma_x^{(2)} + \Sigma_x^{(2)}A) = \nu\beta(1 - \nu\beta)(\Sigma_{dx}^{(1)}A + A\Sigma_{xd}^{(1)}) + (1 - \nu\beta)^2 A\Sigma_x^{(1)}A. \tag{67}$$

Plugging (64) into (65) and (66), we obtain

$$(\Sigma_{xd}^{(2)} + \Sigma_{dx}^{(2)}) - (A\Sigma_x^{(2)} + \Sigma_x^{(2)}A) = -\frac{2\beta(1 + \nu + \beta - \nu\beta)}{1 - \beta^2}(\Sigma_{dx}^{(1)}A + A\Sigma_{xd}^{(1)}) - \frac{4(1 + \beta - \nu\beta)}{1 + \beta}A\Sigma_x^{(1)}A. \tag{68}$$

Plugging (64) into (67), we obtain

$$\nu\beta(\Sigma_{xd}^{(2)}+\Sigma_{dx}^{(2)})+(1-\nu\beta)(A\Sigma_x^{(2)}+\Sigma_x^{(2)}A) = \frac{2\nu\beta(1+\beta-\nu\beta)}{1+\beta}(\Sigma_{dx}^{(1)}A+A\Sigma_{xd}^{(1)})+2(1-2\nu\beta+\frac{2\nu^2\beta^2}{1+\beta})A\Sigma_x^{(1)}A. \tag{69}$$

Combining (68) and (69), we obtain

$$A\Sigma_x^{(2)} + \Sigma_x^{(2)}A = \frac{2\nu\beta}{1-\beta}(\Sigma_{dx}^{(1)}A + A\Sigma_{xd}^{(1)}) + 2A\Sigma_x^{(1)}A. \tag{70}$$

Let's get an expression for $\mathbf{tr}(A\Sigma_\xi)$. From (70) we get (by taking trace and dividing by 2)

$$\mathbf{tr}(A\Sigma_x^{(2)}) = \frac{2\nu\beta}{1-\beta}\mathbf{tr}(\Sigma_{dx}^{(1)}A) + \mathbf{tr}(A\Sigma_x^{(1)}A) \tag{71}$$

From (62) we get (by multiplying by A, taking trace and dividing by 2)

$$\mathbf{tr}(\Sigma_{dx}^{(1)}A) = \mathbf{tr}(A\Sigma_x^{(1)}A) - \frac{(1+\beta-\nu\beta)}{1+\beta}\mathbf{tr}(A\Sigma_\xi) \tag{72}$$

Plugging (72) into (71) we get

$$\begin{aligned}\mathbf{tr}(A\Sigma_x^{(2)}) &= \frac{2\nu\beta}{1-\beta}\left(\mathbf{tr}(A\Sigma_x^{(1)}A) - \frac{(1+\beta-\nu\beta)}{1+\beta}\mathbf{tr}(A\Sigma_\xi)\right) + \mathbf{tr}(A\Sigma_x^{(1)}A) \\ &= \left(\frac{2\nu\beta}{1-\beta} + 1\right)\mathbf{tr}(A\Sigma_x^{(1)}A) - \frac{2\nu\beta}{1-\beta}\frac{(1+\beta-\nu\beta)}{1+\beta}\mathbf{tr}(A\Sigma_\xi)\end{aligned} \tag{73}$$

From (63) we get (by multiplying by A and taking trace)

$$\mathbf{tr}(A\Sigma_x^{(1)}A) = \frac{1}{2}\mathbf{tr}(A\Sigma_\xi) \tag{74}$$

and also by taking trace

$$\mathbf{tr}(A\Sigma_x^{(1)}) = \frac{1}{2}\mathbf{tr}(\Sigma_\xi) \tag{75}$$

Finally we get

$$\begin{aligned}\mathbf{tr}(A\Sigma_x) &= \alpha\mathbf{tr}(A\Sigma_x^{(1)}) + \frac{\alpha^2}{2}\mathbf{tr}(A\Sigma_x^{(2)}) + O(\alpha^3) \\ &= \frac{\alpha}{2}\mathbf{tr}(\Sigma_\xi) + \frac{\alpha^2}{2}\left[\left(\frac{2\nu\beta}{1-\beta} + 1\right)\mathbf{tr}(A\Sigma_x^{(1)}A) - \frac{2\nu\beta}{1-\beta}\frac{(1+\beta-\nu\beta)}{1+\beta}\mathbf{tr}(A\Sigma_\xi)\right] + O(\alpha^3) \\ &= \frac{\alpha}{2}\mathbf{tr}(\Sigma_\xi) + \frac{\alpha^2}{2}\left[\left(\frac{2\nu\beta}{1-\beta} + 1\right)\frac{1}{2}\mathbf{tr}(A\Sigma_\xi) - \frac{2\nu\beta}{1-\beta}\frac{(1+\beta-\nu\beta)}{1+\beta}\mathbf{tr}(A\Sigma_\xi)\right] + O(\alpha^3) \\ &= \frac{\alpha}{2}\mathbf{tr}(\Sigma_\xi) + \frac{\alpha^2}{4}\left[\left(\frac{2\nu\beta}{1-\beta} + 1\right) - \frac{4\nu\beta}{1-\beta}\frac{(1+\beta-\nu\beta)}{1+\beta}\right]\mathbf{tr}(A\Sigma_\xi) + O(\alpha^3) \\ &= \frac{\alpha}{2}\mathbf{tr}(\Sigma_\xi) + \frac{\alpha^2}{4}\mathbf{tr}(A\Sigma_\xi) + \frac{\alpha^2}{4}\frac{2\nu\beta}{1-\beta}\left[1 - \frac{2(1+\beta-\nu\beta)}{1+\beta}\right]\mathbf{tr}(A\Sigma_\xi) + O(\alpha^3) \\ &= \frac{\alpha}{2}\mathbf{tr}(\Sigma_\xi) + \frac{\alpha^2}{4}\mathbf{tr}(A\Sigma_\xi) + \frac{\alpha^2}{4}\frac{2\nu\beta}{1-\beta}\left[\frac{2\nu\beta}{1+\beta} - 1\right]\mathbf{tr}(A\Sigma_\xi) + O(\alpha^3)\end{aligned}$$

The special cases of SGD, SHB and NAG can be straightforwardly obtained by substituting corresponding value of $\nu$ into the general formula.

$\square$

Figure 6: Approximation error of equation (15). Relative error is shown with color and we threshold it at 0.2.

# F   Evaluation of Stationary Distribution Size

In this section we describe experimental details for evaluation of stationary distribution size on different machine learning problems (Section 5). The first problem we consider is a simple 2-dimensional quadratic function, where we add additive zero-mean Gaussian noise independent of the point $x$, so that all assumptions of Theorem 5 are fully satisfied. For this function we have

$$\mu = 0.1, L = 10.0, \Sigma_\xi = \begin{bmatrix} 0.3 & 0 \\ 0 & 0.3 \end{bmatrix}$$

We run the QHM algorithm for 1000 iterations starting at the optimal value and plot final loss as average across all 1000 iterations. We evaluate QHM for the following sweeps of hyperparameters: 30 values of $\alpha$ on a uniform grid on $[0.01, 1.5]$, 30 values of $\beta$ on a uniform grid on $[0, 0.999]$, 30 values of $\nu$ on a uniform grid on $[0, 1]$. For each combination of hyperparameters we verify that $\alpha \to 1, \beta \to 1$ indeed decreases the average loss. However, for smaller values of $\nu$ the effect of $\beta$ is smaller, as expected. The dependence on $\nu$ can be described by a quadratic function with minimum at some $\nu < 1$. Note that from formula (15) the dependence on $\nu$ is indeed quadratic with optimal $\nu$ given by

$$\nu_*(\beta) = \begin{cases} \frac{1+\beta}{4\beta} & , \frac{1}{3} \le \beta < 1 \\ 1 & , 0 \le \beta < \frac{1}{3} \end{cases} \tag{76}$$

From this equation the optimal $\nu_*(\beta) \ge 0.5$ and $\nu_*(\beta) \to 0.5$ as $\beta \to 1$. In the experiments we see the same qualitative behavior, but the optimal value of $\nu$ is much closer to 1 than predicted by equation (76).

The second problem we consider is logistic regression on MNIST dataset. We run QHM for 50 epochs with batch size of 128 and weight decay (applied both to weights and biases) of $10^{-4}$ (thus, $\mu \approx 10^{-4}$). The final loss is averaged across last 1000 batches. We evaluate algorithm for 50 values of $\alpha \in [0.01, 30]$ (log-uniform grid), 20 values of $\beta \in [0, 0.999]$ (uniform grid), 20 values of $\nu \in [0, 1]$ (uniform grid).

The final problem we consider is ResNet-18 on CIFAR-10 dataset. We run QHM with batch size of 256 and weight decay of $10^{-4}$ (applied only to weights). We run algorithm for 80 epochs with constant parameters and average final loss across last 100 batches. We evaluate $\alpha \in \{0.01, 0.05, 0.1, 0.5, 1.0, 2.0, 3.0, 5.0, 7.0, 8.5\}$, $\beta \in \{0.0, 0.01, 0.2, 0.5, 0.7, 0.9, 0.99, 0.999\}$. In this experiment we always set $\nu = 1$.

# G   Approximation error of Theorem 5

In this section we run a set of experiments to check for which values of parameters the equation (15) is not accurate. In fact, we can immediately see that the approximation error grows unboundedly as $\beta \to 1$ if $\nu \notin \{0, \beta, 1\}$, because the right-hand-side of equation (15) converges to $-\infty$, while the left-hand-side is bounded from below.

Since we are interested in the approximation error from the higher-order terms, we run experiments on a 2-dimensional quadratic problem where all assumptions are satisfied. We follow the same experimental settings as in the appendix F. We test a uniform grid of 20 $\beta$ and 20 $\nu$ values on $[0, 1]$ for

$\alpha \in \{0.05, 0.1, 0.2\}$. Note, that we can compute the right-hand-side of equation (15) exactly, but need to estimate the left-hand-side. For that we run QHM for 10000 iterations and compute an empirical covariance of the iterates. Figure 6 shows the results of this experiment. We plot a relative error with color and threshold it at 0.2 (i.e. we consider the formula to be inaccurate if the relative difference between right-hand-side and left-hand-side is bigger than 20%). We can see that indeed when $\alpha$ is moderately big, the formula becomes imprecise for many different values of $\nu$ and $\beta$. However, when $\alpha$ is small, the formula is only imprecise for a very large values of $\beta$ and it becomes more inaccurate when $\nu$ is far from 0 or 1.

## Footnotes

[4]Note that we change the notation to be consistent with the notation of QHM.

[5]In fact, for non-constant $\beta_k$ the algorithms are no longer equivalent.

[6]See, e.g. [30] for the exact form of matrix $P$.