[Reviews · NeurIPS 2019]

Reviewer 1



In this paper, the authors use a general formulation of QHM and derive a unified analysis for several popular methods. Originality: The topic of the paper, in my opinion, is interesting. The paper presents analysis and insights for several methods that are used in practice without theoretical guarantees. Quality: The overall quality of the paper is very good. The paper is a purely theoretical paper. Some parts of the paper are hard to follow and some of the theoretical results are hard to parse (e.g., Theorem 3). Clarity: The paper is well-written and motivated. Significance: In my opinion, even though the paper is purely theoretical, it does provide some insights that may aid in the use of stochastic momentum methods in practice. Issues/Questions/Comments/Suggestions: - Contributions: This section can be written more clearly. And the importance of the results can be better explained. - Contributions: “stochastic noise” what do the authors mean by this? Stochastic noise in the gradient approximations? - Theorem 2: The bounded noise assumption seems like a very strong assumption. The authors should comment about this. On a similar note, it appears that the result that \vu_k \beta_k -> 1 is derived under this strong assumption, whereas the \beta_k -> 0 result is not. The authors should comment about this in the paper. Moreover, why is the \vu_k \beta_k -> 1 result interesting? The authors should comment about this. - Theorem 3: 0<, \mu \leq … Is this correct? - Theorem 3: What is \epsilon_k? - Theorem 3: This result excludes the case of \beta = 1. The authors should comment about this in the paper. - Figures 1, 2 and 3 are very interesting. The authors, in my opinion, should expand the discussion about the results shown in these figures (e.g., the three regimes shown in 1c).

Reviewer 2



INDIVIDUAL COMMENTS / QUESTIONS 1) I really appreciate how the paper ties up loose ends by unifying the analysis of several momentum-based methods in the stochastic setting. 2) I am sorely missing a literature review. I am not very closely familiar with the literature analyzing momentum methods, but there's a lot of work out there (e.g., the line of research studying momentum methods in the continuous time limit). A brief review would be very helpful to position the paper within the existing work. 3) On Section 4: a) You should prominently cite Mandt et al. [16], who show similar results for SGD. b) In the beginning, it says "we use quadratic functions for ease of analysis". To me this implies that the analysis would go through for more general functions. I don't find it obvious that it would. I think this should be justified or the actual more general analysis should be presented. 4) I would recommend the general version of Theorem 3 (i.e., Theorem 6 in the appendix) in the main paper. Specializing to quadratic functions imho does not add any extra insight here, so I would state the more general result. TYPOS / STYLE - Line 70: Section title should be lower case - Line 138: Should be "driven by i.i.d. noise" - Line 193: should be "[...] then the QHM algorithm [...]" - Line 216: should be "visualization" for consistent American English - Line 264: "evidence" does not have a plural form - Consistently capitalize "Section" when referencing specific sections, e.g., line 210 - The references could use some cleaning: Capitalization [e.g. 1, 13, 15, 16]. The URL in [13] RATING Quality: To the best of my knowledge, the paper is mathematically sound, but I followed the proofs in the appendix only superficially due to time restrictions. - Clarity: The paper is well written and all ideas are explained very clearly. I am missing a brief literature review to set the scene and position the paper in the existing work. - Significance: Momentum methods are widely used but not very well-understood in the stochastic setting. - Originality: This is not a paper with grand new ideas, but it is tying up some loose ends by unifying the analysis of several momentum-based methods. - Reproducibility: No code has been provided and the description of the experimental setup is, in my opinion, insufficient for an outsider to reproduce the results. Overall, this is a very solid paper and I recommend acceptance.

Reviewer 3



The paper partially fills the blank of the theoretical analysis of the original QHM paper and is well written. Since the momentum based SGD is frequently used in deep learning, the proposed theories could help one tune parameters. I vote for acceptance.

[Author Response · NeurIPS 2019]

We thank the reviewers for their time, effort, and helpful feedback. We will make sure to correct all typos and incorporate
the reviewers' suggestions in the final version of the paper. We address individual feedback below.

**Reviewer 1:**

Yes, "stochastic noise" is the error in the gradient approximations, i.e. $\nabla F(x^k) - g^k$ in the notation of the paper. We
will add this definition to the paper.

Bounded noise is indeed a strong assumption, and we are not sure whether the proof can be extended to the more
general case where $\mathbf{E}_k[||\xi^k||^2]$ is uniformly bounded. We should note that [25] proves a similar result for SGM with a
more general noise condition than bounded noise, and their technique may extend to QHM, but bounded noise greatly
simplifies the proof. We will comment on this in the paper. In our opinion, the result showing convergence of QHM
when $\nu_k\beta_k \to 1$ is interesting from both theoretical and practical perspectives. From the theoretical side, the result
shows that it is possible to always be increasing the amount of momentum (in the limit when $\nu_k\beta_k = 1$, we are not
using the fresh gradient information at all!) and still obtain convergence (which is not too surprising, since the objective
function is smooth). From the practical point of view, Theorem 5 shows that increasing $\nu_k\beta_k \to 1$ might lead to smaller
stationary distribution size, which may give better empirical results. We will include this discussion in the final version
of the paper.

Theorem 3 guarantees that there exists some sequence $\epsilon_k$ such that the statement holds. We will update the statement of
Theorem 3 to include $\mu > 0$ and $\exists\{\epsilon_k\}$ with $\epsilon_k \geq 0$. When $\beta = 1$ and is constant, the algorithm cannot converge, since
the gradient information is not used at all. However, as we show in Section 2, it is possible to set $\beta_k \to 1$ in the limit
and still obtain convergence. As mentioned above, we found this difference theoretically interesting and will highlight
it after Theorem 3.

We had to limit the explanation of Figures 1, 2, and 3 due to space constraints. We would use the additional page
allowed in the final version to include more interpretation of these figures, explaining their relationships to Theorems 3,
4, and 5 in more detail.

To improve the presentation of our theoretical results, we will include more lead-in discussion before our theorems
(especially Theorem 3) to make them easier to understand. We believe that combining our literature review into its own
section (as suggested by Reviewer 2 and addressed below) will also help the reader better contextualize our theorems.
Regarding sensitivity, we will expand Section 5 with comments that the presented results depend continuously on the
algorithm parameters.

**Reviewer 2:**

Since we presented a range of results covering different aspects of momentum methods (asymptotic convergence,
stability, and stationary distributions), and each of these areas has its own rich literature, we put a brief review of the
relevant works at the beginning of each section. To improve readability, we will combine these references in a separate
section to give a narrative that ties the different areas together.

We cite Mandt et al. [16] for some empirical justification of our Assumption A3 (in the beginning of Section 3) and for
proving an equivalent result to our Theorem 4 in the case of stochastic heavy ball (in the beginning of Section 4). We
can make the existing citation after Theorem 4 more prominent.

By saying that "we use quadratic functions for ease of analysis" we did not mean to imply that it is straightforward to
prove our results for a more general class of functions, although the basic principles remain similar. We will change the
wording of this phrase in the final version of the paper to make it more specific. The results in Theorems 4 and 5 may
hold approximately near a local minimum for nonconvex but sufficiently smooth functions, when the learning rate is
sufficiently small. In that case, the function is intuitively almost quadratic locally, since the higher-order terms in the
Taylor expansion around the optimum can be neglected.

For most of our experiments, the code is very minimal. To make it easier to reproduce our results, we will release the
code for all of the experiments performed in this paper.

**Reviewer 3:**

We agree with the reviewer that a more general analysis is necessary to better understand the properties of momentum
methods in the wild, especially for the nonconvex problems faced in deep learning practice. While our results are based
on some restrictive assumptions, we hope they still provide valuable practical insights, and we believe they could serve
as an important foundation for any future work extending the unified analysis of momentum methods to more general
classes of functions.

[Meta-Review · NeurIPS 2019]

The reviewers agree that the topic tackled in the paper is interesting and the mathematical results are promising. Overall, this submission is a good attempt in deriving a mathematical understanding of QHM, but the results are often only partially investigated and commented. For instance, in section 3 the main result (i.e. the convergence rate for quadratics) is really hard to parse and is poorly commented in the sense that its practical value is unclear. The paper also makes a number of conjectures that are not backed up and the authors are therefore advised to tone down their claims. This includes "we conjecture that the optimal convergence rate is a monotonically decreasing function of nu" as well as the quality of the approximation in Section 4. In conclusion, all three reviewers liked the paper but also highlighted some shortcomings, therefore justifying acceptance as a poster but not an oral.